# Dynamic molecular evolution of a supergene with suppressed recombination in white-throated sparrows

**Hyeonsoo Jeong[1†], Nicole M Baran[1,2,3†], Dan Sun[1,4], Paramita Chatterjee[1], Thomas S Layman[1], Christopher N Balakrishnan[5], Donna L Maney[2]\*, Soojin V Yi[1,3]\***

[1]School of Biological Sciences, Georgia Institute of Technology, Atlanta, United States; [2]Department of Psychology, Emory University, Atlanta, United States; [3]Department of Ecology, Evolution, Marine Biology, University of California, Santa Barbara, Santa Barbara, United States; [4]Department of Medicine Huddinge, Karolinska Institutet, Stockholm, Sweden; [5]Department of Biology, East Carolina University, Greenville, United States

**Abstract** In white-throated sparrows, two alternative morphs differing in plumage and behavior segregate with a large chromosomal rearrangement. As with sex chromosomes such as the mammalian Y, the rearranged version of chromosome two (ZAL2$^m$) is in a near-constant state of heterozygosity, offering opportunities to investigate both degenerative and selective processes during the early evolutionary stages of 'supergenes.' Here, we generated, synthesized, and analyzed extensive genome-scale data to better understand the forces shaping the evolution of the ZAL2 and ZAL2$^m$ chromosomes in this species. We found that features of ZAL2$^m$ are consistent with substantially reduced recombination and low levels of degeneration. We also found evidence that selective sweeps took place both on ZAL2$^m$ and its standard counterpart, ZAL2, after the rearrangement event. Signatures of positive selection were associated with allelic bias in gene expression, suggesting that antagonistic selection has operated on gene regulation. Finally, we discovered a region exhibiting long-range haplotypes inside the rearrangement on ZAL2$^m$. These haplotypes appear to have been maintained by balancing selection, retaining genetic diversity within the supergene. Together, our analyses illuminate mechanisms contributing to the evolution of a young chromosomal polymorphism, revealing complex selective processes acting concurrently with genetic degeneration to drive the evolution of supergenes.

**\*For correspondence:**
dmaney@emory.edu (DLM);
soojinyi@ucsb.edu (SVY)

[†]These authors contributed equally to this work

**Competing interest:** The authors declare that no competing interests exist.

## Editor's evaluation

In this important paper, the authors generate and analyze new genome and gene expression data to understand better the evolution of the white-throated sparrow supergene region, which contains 1000 genes and determines whether a bird has a tan or a white stripe. The study convincingly illustrates how the cessation of recombination that results from a chromosomal inversion can become a source of evolutionary novelty. The lack of recombination can result in the accumulation of deleterious variation leading to degeneration, but it can also (as here) facilitate genomic diversification and adaptation. The results will be of interest to a broad array of researchers studying genome architecture and phenotypic diversity and evolution.

## Introduction

Supergenes comprise closely linked genetic variants that are maintained due to suppressed recombination (*Charlesworth, 2016*; *Thompson and Jiggins, 2014*). Their evolution presents an interesting paradox, in that the suppression of recombination that occurs inside supergenes reduces the efficacy of natural selection, leading to genetic degeneration. At the same time, supergenes are associated with dramatically divergent, adaptive phenotypes. These divergent phenotypes, which include classic examples of Batesian mimicry and self-incompatibility in flowering plants (*Charlesworth, 2016*; *Thompson and Jiggins, 2014*; *Otto and Lenormand, 2002*) and striking polymorphisms in social behavior (*Wang et al., 2013*; *Huang et al., 2018*; *Yan et al., 2020*; *Martinez-Ruiz et al., 2020*; *Farrell et al., 2013*; *Küpper et al., 2016*; *Lamichhaney et al., 2016*), have long inspired both theoretical and empirical studies of their evolution. Recent genome-scale studies have illuminated wide-ranging impacts of supergene evolution on complex phenotypes across diverse taxa (e.g. *Schwander et al., 2014*; *Pearse et al., 2019*; *Hager et al., 2022*; *Joron et al., 2011*; *Kunte et al., 2014*; *Kess et al., 2019*; *Lundberg et al., 2017*; *Roberts et al., 2009*; *Sanchez-Donoso et al., 2022*; *Funk et al., 2021*). Currently, the mechanisms by which functionally divergent supergene haplotypes evolve in the face of multiple evolutionary forces remain poorly understood, presenting a critical gap in knowledge.

One notable example of a supergene associated with social behavior is found in white-throated sparrows (*Zonotrichia albicollis*), in which a large supergene co-segregates with parental behavior and aggression (*Tuttle, 2003*; *Maney et al., 2015*; *Horton et al., 2013*; *Tuttle et al., 2016*; *Sun et al., 2018*; *Merritt et al., 2020*). White-throated sparrows occur in two alternative plumage morphs, white- and tan-striped (*Lowther, 1961*). These morphs differ not only in their plumage coloration, but also in their social behavior, with white-striped birds exhibiting increased aggression and more frequent extra-pair copulations, and tan-striped birds engaging in more parental care compared with birds of the white-striped morph (*Tuttle, 2003*; *Maney et al., 2015*; *Maney, 2008*; *Horton et al., 2012*). These alternative morphs are linked to a large (~100 Mbp, >1 k genes) rearrangement on the second largest chromosome, called ZAL2$^m$, so named because the rearranged chromosome is metacentric. White-striped birds are heterozygous for ZAL2$^m$ and the sub-metacentric chromosomal arrangement, ZAL2, whereas tan-striped birds are homozygous for ZAL2 (30, 31).

In addition to highly divergent social behavior, this relatively young supergene (estimated to have arisen 2–3 million years ago *Tuttle et al., 2016*; *Thomas et al., 2008*; *Huynh et al., 2010*), is also associated with a remarkable disassortative mating system that maintains the 'balanced' morph frequencies in the population. Almost all breeding pairs consist of one bird of each morph, earning the species the moniker 'the bird with four sexes' (*Campagna, 2016*). Breeding pairs consisting of two individuals of the same morph are estimated to occur less than 1% of the time (*Tuttle et al., 2016*; *Thorneycroft, 1966*) and only six ZAL2$^m$ homozygotes (i.e. 'super-white' birds) have ever been identified (*Horton et al., 2013*; *Tuttle et al., 2016*; *Thorneycroft, 1975*; *Falls and Kopachena, 2020*) out of thousands of birds karyotyped or genotyped. Given that ZAL2$^m$ exists in a near-constant state of heterozygosity, it is in a state of suppressed recombination, similar to the Y and W sex chromosomes in mammals and birds, respectively. The suppression of recombination on ZAL2$^m$ is expected to reduce the efficacy of natural selection, leading to reduced genetic diversity and the degeneration of the chromosome (*Barton and Charlesworth, 1998*; *Charlesworth, 2012*). On the other hand, the tight linkage of alleles within the ZAL2$^m$ supergene may contribute to adaptive phenotypes (*Tuttle et al., 2016*; *Sun et al., 2018*; *Maney et al., 2020*). Therefore, this system provides a unique opportunity to investigate the evolution of a supergene underlying social and mating behavior (*Tuttle et al., 2016*; *Sun et al., 2018*; *Merritt et al., 2020*; *Sun et al., 2021*).

Here we aim to better understand the evolutionary forces shaping the ZAL2 and ZAL2$^m$ chromosomes. Our goal was to address two unanswered questions. First, to what extent has ZAL2$^m$ degenerated? Early analyses of the rearrangement (*Davis et al., 2011*) did not show signals of degeneration, such as pseudogenization or the accumulation of repetitive sequences. However, *Tuttle et al., 2016* found a weak signal of excess non-synonymous polymorphism for genes inside the rearranged region on ZAL2$^m$ and reduced allelic expression for ZAL2$^m$ genes, which could be consistent with functional degradation of ZAL2$^m$ (*Tuttle et al., 2016*). (*Sun et al., 2018*) similarly found a slightly higher number of non-synonymous substitutions and an increased ratio of non-synonymous to synonymous substitution rates ($d_N/d_S$) on ZAL2$^m$ compared with ZAL2. (*Sun et al., 2018*) also found reduced expression of ZAL2$^m$ alleles in brain tissue, perhaps suggesting that the accumulation of deleterious mutations has

led to reduced expression of genes from ZAL2$^m$. Their additional finding of reduced accumulation of mutations in functional regions suggested that ZAL2$^m$ has, in fact, experienced purifying selection to remove deleterious alleles. Thus, while there is some evidence that ZAL2$^m$ has degenerated, these results have been inconsistent and somewhat inconclusive.

Second, what are the selective forces shaping the genomic landscapes of both ZAL2 and the ZAL2$^m$ supergene? The signals of both purifying and positive selection have been relatively weak in previous genomic analyses of ZAL2 and ZAL2$^m$ (*Tuttle et al., 2016*; *Sun et al., 2018*). Yet, by definition, ZAL2$^m$ must contain variation that underlies the differences between the white- and tan-striped morphs (*Fisher, 1931*; *Bull, 1983*; *Charlesworth and Charlesworth, 1980*; *Rice, 1987b*; *Rice, 1987a*). There is already some evidence that this variation affects behavior; allelic differences in the promoter region of the gene encoding estrogen-receptor alpha (ESR1) are likely to alter expression (*Merritt et al., 2020*), and the expression of this gene was shown to be necessary for aggressive behavior typical of the white-striped morph (*Merritt et al., 2020*). ZAL2$^m$ is also associated with differential expression of a key neuromodulator, vasoactive intestinal peptide (*Horton et al., 2020*), known to be causal for aggression in songbirds (*Goodson et al., 2012*).

Investigations of young heteromorphic sex chromosomes suggest that the accumulation of sexually antagonistic genes (i.e. genes that are beneficial to one sex and harmful to the other) may in fact drive the evolution of sex chromosomes (*Bachtrog, 2004*; *Bachtrog, 2006*; *Zhou and Bachtrog, 2012*). For example, positive selection at a small number of antagonistic alleles was shown to be a potent force shaping evolution of the young Y chromosomes in *Drosophila miranda* (*Bachtrog, 2004*) even in the face of degeneration of other genes elsewhere on the chromosome. In white-throated sparrows, evidence of positive selection on both ZAL2 and ZAL2$^m$ has been quite limited (*Tuttle et al., 2016*; *Sun et al., 2018*). Nonetheless, the discovery of ZAL2- and ZAL2$^m$-specific alleles that benefit the tan- and white-striped morphs, respectively (*Merritt et al., 2020*; *Horton et al., 2020*), suggests that antagonistic selection likewise contributes to the evolution of both ZAL2 and ZAL2$^m$.

In addition to antagonistic selection, balancing selection may be implicated in the evolution of ZAL2$^m$. The negative assortative mating system in white-throated sparrows, which maintains the chromosomal polymorphism, is a canonical example of balancing selection (*Huynh et al., 2010*). However, balancing selection is also a way of maintaining advantageous genetic diversity in populations, which may be especially critical in the context of a non-recombining chromosome. Indeed, balancing selection appears to be more common in self-fertilizing (selfing) vs non-selfing species, which are likewise characterized by reduced genetic diversity, increased linkage disequilibrium, and reduced efficacy of selection (*Glémin, 2021*; *Glémin et al., 2019*; *Delph and Kelly, 2014*; *Gaut et al., 2015*). Therefore, balancing selection may maintain multiple alleles inside non-recombining regions of chromosomes like ZAL2$^m$.

Previous studies have been limited in the extent to which they could test directly for degeneration, adaptive changes on ZAL2$^m$, and selection at the genome level. These limitations stemmed from low sample sizes of sequencing data, the reduced intraspecies variability, and a low-quality ZAL2$^m$ assembly that prevented detection of long-range haplotypes (*Tuttle et al., 2016*; *Sun et al., 2018*; *Thomas et al., 2008*). Here, we overcome these challenges by analyzing extensive genomic, transcriptomic, and population data, providing insight into the evolution of young supergenes.

## Results

### Novel and extensive genomic and population data from white-throated sparrows

To better understand the evolutionary history of the ZAL2$^m$ chromosomal rearrangement, we generated additional sequence data from a rare, 'super-white' (ZAL2$^m$ homozygote) bird (*Horton et al., 2013*; *Sun et al., 2018*). We generated variable fragment size libraries consisting of 150 bp paired-end reads (insert size of 300 bp and 500 bp) and 125 bp mate pair reads (insert size of 1 kb, 4–7 kb, 7–10 kb, and 10–15 kb). We performed whole-genome sequencing of an additional 62 birds (49 white-striped birds and 13 tan-striped birds sampled from a variety of locations around the U.S.) (Materials and methods, *Supplementary file 1*). White-striped birds, which are heterozygous for the rearrangement (ZAL2/2$^m$), were sequenced at higher coverage than tan-striped birds (ZAL2 homozygotes) so that we could obtain sufficient reads to separate ZAL2 and ZAL2$^m$ alleles in white-striped

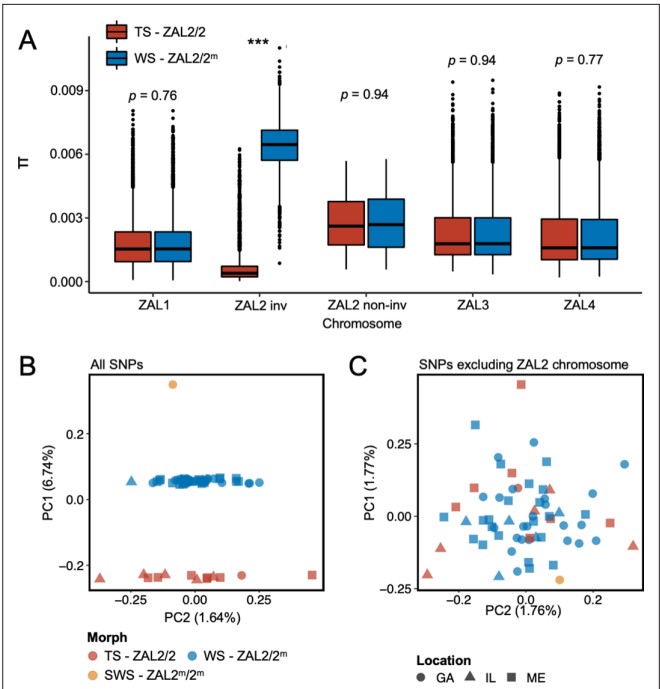

**Figure 1.** Genomic data from newly sequenced tan- and white-striped birds. (**A**) Nucleotide diversity of macro-chromosomes for tan-striped (TS) and white-striped (WS) birds. White-striped birds (ZAL2/2$^m$) show elevated nucleotide diversity for the ZAL2/2$^m$ inverted (INV, i.e. rearranged) regions (ZAL2/2$^m$ inv), while TS birds (ZAL2/2) show overall reduced nucleotide diversity for the inverted regions compared with other chromosomes. Note that panel (**A**) shows the comparison across morph. The comparison across the ZAL2 and ZAL2$^m$ alleles is shown in *Figure 2a*. (**B**) Scatterplots of eigenvector 1 (PC1) and eigenvector 2 (PC2) from principal component analysis of all single-nucleotide variants (left panel). (**C**) Principal component analysis (PCA) excluding single nucleotide polymorphisms (SNPs) on the ZAL2 chromosomes (right panel). The sex chromosomes and the ZAL3 chromosome (which includes an additional chromosomal inversion) were excluded from both PCA analyses. Note that 'location' here refers to the site of collection or capture of the bird: Georgia (GA), Illinois (IL), or Maine (ME). Breeding locations for GA and IL birds are unknown.

The online version of this article includes the following source data and figure supplement(s) for figure 1:

**Source data 1.** Nucleotide diversity between tan- and white-striped birds.

**Figure supplement 1.** The number of informative sites inside the ZAL2$^m$ rearrangement differed between morphs.

**Figure supplement 2.** Admixture tests showed no population substructure by geographic sampling location.

individuals (average mean depth coverages were 41.5 × vs 28.4 × for white- and tan-striped birds, respectively, *Supplementary file 2*). Genomic variants were called according to the guidelines of Genome Analysis Toolkit (GATK) (ver. 4.1) (Materials and methods), leading to the discovery of a total of 11,382,994 single nucleotide polymorphisms (SNPs). None of the samples showed evidence of family relationships when we computed relatedness estimates between individuals. Consequently, we used all samples in the subsequent analyses. We found a significantly higher number of polymorphic sites within white-striped birds than tan-striped birds exclusively for ZAL2/2$^m$ chromosomal regions (*Figure 1—figure supplement 1*). Nucleotide diversity of the rearranged region of the ZAL2/2$^m$ chromosomes was elevated in white-striped birds compared with tan-striped birds, suggesting distinctive patterns between the two plumage morphs (*Figure 1a*).

Among the total SNPs identified, 12.6% (N=1,439,991) resided on scaffolds we have previously assigned to the ZAL2/2$^m$ chromosome (*Sun et al., 2018*). Principal component analysis (PCA) of these ZAL2/2$^m$ SNPs revealed distinct clusters corresponding to the morphs (*Figure 1b*). The first principal component (PC1), which explained 6.7% of the variation in the data, clearly separated tan- and white-striped birds, with the lone super-white individual (ZAL2$^m$/2$^m$ homozygote) as a clear outlier. In contrast, other available phenotypic information, including sex and geographic origin of samples, did not show meaningful variation with the principal components, and other PCs had little explanatory

power (*Figure 1b*). Tests for admixture also failed to identify significant population substructures by geographical origin of samples (*Figure 1—figure supplement 2*). This lack of population structure is unsurprising, as 35 of the 63 samples (56%) were from birds that were migrating, and, thus, the breeding location of these birds is unknown.

## Features of the ZAL2$^m$ chromosome consistent with reduced efficacy of natural selection and low levels of recombination

We examined several genomic features of the ZAL2$^m$ chromosome using the additional genomic resources we generated. We first performed a *de novo* genome assembly of the super-white bird, employing newly generated sequence data, to study the ZAL2$^m$ chromosome with an assembly derived entirely from a bird homozygous for the ZAL2$^m$ chromosome. The total assembly size was 1058 Mbp (N50 length of 3.1 Mbp, longest scaffold 27 Mbps), comparable to that of the ZAL2/2 reference assembly (1052 Mbp, N50 scaffold length of 4.86 Mbp, longest scaffold 45 Mbp) (see *Supplementary file 3* for more details). There were 160 putatively ZAL2$^m$-linked scaffolds (Materials and methods), with a total length (110.99 Mbp) comparable with that of ZAL2-linked scaffolds from the reference assembly (108.5 Mbp *Tuttle et al., 2016*). Despite this similarity in total length, however, the average length of the individual ZAL2$^m$-linked scaffolds was significantly shorter than scaffolds on other chromosomes in the super-white assembly (p<0.001, Mann-Whitney U-test). It was also shorter than the average scaffold length on the ZAL2 chromosome in the ZAL2/2 reference assembly (*Figure 2a*). We did not observe such a pattern in the other chromosomes of similar size when comparing between the two assemblies (*Figure 2a*). This result was consistent with the presence of repetitive DNA sequences on ZAL2$^m$ causing more assembly breaks compared with the ZAL2/2 reference genome. We found evidence that the ZAL2$^m$ chromosome contained more repeat elements and was especially enriched for long terminal repeat elements (2.4 Mbp vs 2.1 Mbp) and interspersed repeats (5.8 Mbp vs 5.5 Mbp), compared with the ZAL2 chromosome. The number of these repeat elements is likely to be underestimated, given that the ZAL2$^m$ assembly is highly fragmented. Additionally, we found that ZAL2 and ZAL2$^m$ had accumulated a higher proportion of structural variants (insertions and deletions) compared with other chromosomes (*Figure 2b*).

Next, using the large amount of newly generated population genomic data, we examined patterns of SNPs on ZAL2$^m$ alleles separately from those on ZAL2 via haplotype phasing using fixed differences between the two chromosome types (Materials and methods). We found that the total number of genetic variants was approximately 12-fold reduced on the ZAL2$^m$ alleles compared with ZAL2 alleles (29,420 vs 367,466 SNPs and 3,921 vs 32,479 indels on ZAL2$^m$ and ZAL2, respectively, after excluding singleton variants). The mean nucleotide diversity was similarly reduced on the ZAL2$^m$ chromosome compared with the ZAL2 chromosome (0.0104% vs 0.1141% for ZAL2$^m$ vs ZAL2, respectively).

Analyses of the genetic variants on the ZAL2 and ZAL2$^m$ alleles showed evidence of only weak genetic degeneration. The ratio of non-synonymous to synonymous fixed differences inside the rearranged region was slightly, but significantly, elevated for ZAL2$^m$-derived compared with ZAL2-derived fixed differences (*Figure 2c*), which is consistent with either positive selection or inefficient purifying selection on ZAL2$^m$. We found that the ratio of non-synonymous to synonymous nucleotide diversity ($\pi_N/\pi_S$) was significantly increased on ZAL2$^m$ compared with ZAL2 (p=1.3 × 10$^{-13}$, Mann-Whitney U-test, *Figure 2d*). The minor allele site frequency spectrum (SFS) for the ZAL2$^m$ synonymous and non-synonymous sites showed a large proportion of singleton variants and an irregular decay of allele frequency as the minor allele count increases (*Figure 2e*), also suggesting reduced efficacy of purifying selection on ZAL2$^m$.

Assuming the mutation rates of the two chromosomes are similar, the ratio of effective population size ($N_e$) between the ZAL2$^m$ and ZAL2 can be approximated by the ratio of nucleotide diversity of synonymous sites between the ZAL2 and ZAL2$^m$. The proportion of $N_e$ between ZAL2$^m$ and ZAL2 is 0.12 (±0.01), which is threefold lower than the expected ratio of 0.33 (because the ZAL2$^m$ chromosome is 1/3 as frequent as the ZAL2, given 'balanced' morph frequencies observed in the wild). This lower proportion suggests that $N_e$ of ZAL2$^m$ has undergone further reduction than expected from the census size in the population, consistent with the effects of reduced recombination. We noted that the linkage disequilibrium between variants on ZAL2$^m$ exhibited the classic decay with distance (*Figure 2f*), indicating at least some level of recombination. Taken together, these results are consistent with reduced, but not entirely eliminated, recombination on the ZAL2$^m$ chromosome.

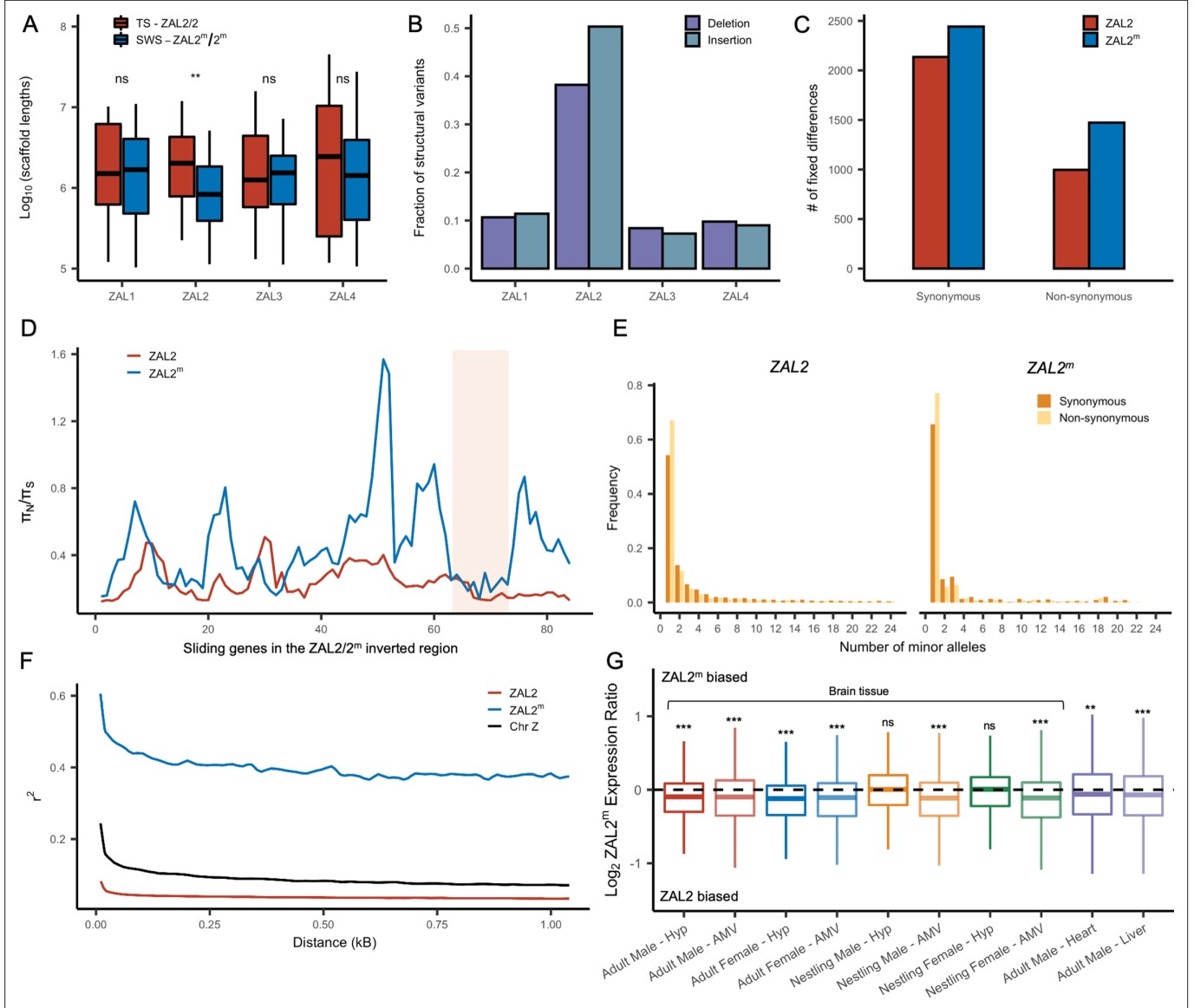

**Figure 2.** Genetic divergence between ZAL2 and ZAL2$^m$ chromosomes. (**A**) The scaffolds for the ZAL2$^m$ chromosome in the super-white (SWS) assembly tend to be fragmented compared with those for the ZAL2 chromosome in the tan-striped (TS) assembly. ** p<0.001 (Mann-Whitney U-test); ns, not significant (**B**) Fraction of structural variants (SV), both insertion and deletion events, for the 4 largest chromosomes, using the tan-striped assembly as a reference. The fraction of SV is computed as a total base affected by variants divided by the length of the chromosome. (**C**) Number of fixed mutations derived in ZAL2 and ZAL2$^m$ in protein-coding regions (**D**) Sliding window (window size of 20 genes with step size of 5 genes) analysis of the ratio of nonsynonymous to synonymous nucleotide diversity ($\pi_N/\pi_S$) within the ZAL2 and ZAL2$^m$ chromosomes. The ZAL2$^m$ outlier region is highlighted (colored background). (**E**) Site frequency spectrum of polymorphic sites. (**F**) Decay of linkage disequilibrium. (**G**) Proportion of the ZAL2$^m$ alleles expressed for each tissue set. The proportion of the ZAL2$^m$ alleles expressed is less than the null hypothesis of 0.5 for all tissues except nestling AMV using false discovery rate (FDR) correction. Hyp, hypothalamus; AMV, ventromedial arcopallium.

The online version of this article includes the following source data and figure supplement(s) for figure 2:

**Source data 1.** Scaffold length.

**Source data 2.** Structural variant proportions.

**Source data 3.** Variant information.

**Source data 4.** Haplotype phased nucleotide diversity data.

**Source data 5.** Minor alleles.

**Source data 6.** Linkage disequilibrium.

**Source data 7.** RNAseq allele specific expression data in long format.

**Figure supplement 1.** Allelic bias in expression was associated with the number of non-synonymous fixed differences.

**Table 1.** List of RNA sequencing data sets.

| | Tissue | Sample size (WS/total) | Collection details | Source |
|---|---|---|---|---|
| | | | | *Zinzow-Kramer et al., 2015*; *Sun et al., 2018* |
| Adult males | Brain (Hyp, AMV) | 9/20 | Collected early in the breeding season | Accession: GSE77186 |
| Adult females | Brain (Hyp, AMV) | 6/11 | Collected early in the breeding season | Accession: PRJNA657006 |
| Nestlings (both sexes) | Brain (Hyp, AMV) | 16/32 | Collected from nests during the breeding season | Accession: PRJNA657006 |
| | | | Collected during fall migration, then housed in captivity on either long or short days to simulate breeding vs non-breeding; collected at two time points during the day | *Horton et al., 2019* |
| Adult males (all white-striped) | Heart and Liver | 20/20 | | Accession: GSE116989 |

We next examined whether degeneration inside the rearranged region on ZAL2$^m$ has resulted in globally reduced expression of the alleles carried by the ZAL2$^m$ supergene variant. To do so, we used multiple large RNAseq datasets from a variety of tissues in birds sampled from different geographic locations and times of year (see Materials and methods, *Table 1*). As predicted and consistent with what was previously reported (*Sun et al., 2018*), we found evidence of consistently reduced expression of the ZAL2$^m$ alleles in 8/10 types of tissue (*Figure 2g*). We next tested for an association between the number of accumulated mutations (non-synonymous, synonymous, and in the promoter region) on ZAL2$^m$ and allelic bias (AB) in expression of the ZAL2$^m$ alleles within each tissue, which would link genetic degeneration within or near genes to reduced expression of ZAL2$^m$. We found evidence that allelic bias in gene expression was associated with the rate of non-synonymous fixed differences ($X^2(1)=9.97$, $p=0.00159$), although the effect size was exceedingly small (marginal $r^2=0.0020$) (*Figure 2—figure supplement 1a*). Neither the rate of synonymous fixed differences ($X^2(1)=1.0098$, $p=0.315$, *Figure 2—figure supplement 1b*) nor the number of fixed differences within 1 kb upstream of the transcription start site ($X^2(1)=0.8992$, $p=0.343$, *Figure 2—figure supplement 1c*) were associated with allelic bias. Thus, the overall reduction in expression of the alleles carried by the ZAL2$^m$ supergene is weakly associated with an increased number of non-synonymous fixed nucleotide changes within genes. The limited and weak nature of the effect suggests, however, that the pattern of gene expression may have been affected also by other factors, for example ongoing selection (thus manifested as nucleotide polymorphism), selection at more distal sites, and/or epigenomic mechanisms, such as differences between ZAL2 and ZAL2$^m$ in DNA methylation or histone modification (see *Sun et al., 2021*).

## Evidence of regional balancing selection on the ZAL2$^m$ chromosome

Although the level of genetic diversity was overall reduced on ZAL2$^m$, it was exceptionally elevated in one region corresponding to ~3 Mbps spanning 5 scaffolds. This region, henceforth referred to as the ZAL2$^m$ outlier region (*Figures 2 and 3*), includes at least 15 protein-coding genes that are well conserved as single copy genes across 13 avian species (*Table 2*). On average, nucleotide diversity in ZAL2$^m$ across this region was 0.001, which is tenfold higher than the mean nucleotide diversity of ZAL2$^m$ and even exceeds the nucleotide diversity in the corresponding region within ZAL2 by 3.15-fold.

To first examine whether this region has recently experienced an introgression event that could have caused the observed patterns, we constructed a phylogenetic tree of this region. The resulting tree exhibited the same topology as those from other regions of the ZAL2$^m$, thus providing no support for introgression (*Figure 3b*). We have not yet resolved the distribution of repetitive elements in the two chromosomes. Thus, to avoid errors in the phylogenetic analysis resulting from the rapid turnover

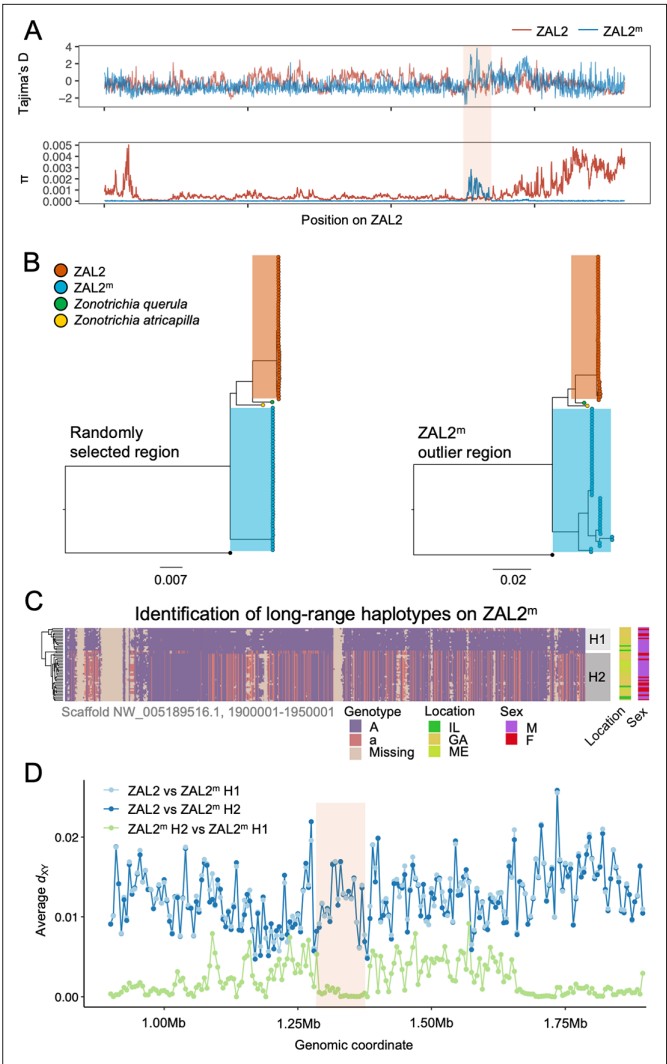

**Figure 3.** Genetic diversity and patterns of divergence across the rearranged region of the ZAL2$^m$ chromosome and in the ZAL2$^m$ outlier region. (**A**) Tajima's D and nucleotide diversity across the ZAL2 and ZAL2$^m$ chromosomes. The ZAL2$^m$ outlier region is highlighted (colored background). (**B**) Phylogenetic tree of randomly selected regions (left panel) and the ZAL2$^m$ outlier region (right panel). The ZAL2$^m$ chromosome shows multiple haplotype structures and has longer branch lengths within the population compared with ZAL2 chromosomes. (**C**) Single nucleotide polymorphism (SNP) genotype plot of a scaffold inside the ZAL2$^m$ outlier region (Scaffold NW_005189516.1, 1900001–1950001). The plot shows two haplogroups. Major allele SNPs (A, same genotype as the super-white ZAL2$^m$/2$^m$ genome) are represented in purple, and minor allele SNPs (a, different from the super-white genome) in red. Tan indicates that there were no fixed SNPs to differentiate ZAL2 vs ZAL2$^m$ reads, resulting in missing data. (**D**) Genetic divergence ($d_{XY}$) for a portion of the rearrangement. $d_{XY}$ between the ZAL2 chromosome and haplogroup 1 (**H1**) is plotted in light blue, between ZAL2 and haplogroup 2 (**H2**) in dark blue, and between H1 and H2 in light green.

The online version of this article includes the following source data and figure supplement(s) for figure 3:

**Source data 1.** RAxML bipartitions for scaffold 5189516.

**Source data 2.** RAxML bipartitions for scaffold 5190802.

**Source data 3.** Genotype data for scaffold 5189516.

**Source data 4.** $d_{XY}$ between ZAL2$^m$ haplotypes and ZAL2.

**Figure supplement 1.** No evidence of introgression in ZAL2$^m$ outlier region.

**Figure supplement 2.** The D-statistic did not vary by haplotype.

**Figure supplement 3.** No difference in sequencing depth between haplotypes.

*Figure 3 continued on next page*

*Figure 3 continued*

**Figure supplement 4.** The ZAL2$^m$ outlier region exhibited an excess of intermediate frequency minor alleles.

**Figure supplement 5.** Neither sex nor geographic location of sample collection produced distinct patterns between haplogroups.

of repetitive elements, we also performed the analysis using only the exons of single-copy orthologous genes, similar to analyses done by *Stolle et al., 2022*. The results (*Figure 3—figure supplement 1*) did not provide support for an introgression event and therefore did not change our conclusions. In addition, we calculated the D-statistic (ABBA-BABA test, which tests for genomic regions that are discordant with the species tree *Martin et al., 2015*) to test for evidence of gene flow from another species from the same genus, Harris' sparrow (*Zonotrichia querula*). Sliding window estimates of the D-statistic did not show any differences in patterns between haplotypes (*Figure 3—figure supplement 2*), suggesting that the multiple haplotype structures in ZAL2$^m$ were not introduced by gene flow from this closely related species or from the ZAL2 chromosome. Nonetheless, the ZAL2$^m$ outlier region showed longer branch lengths than did both the corresponding region on the ZAL2 chromosome and a randomly selected region of ZAL2$^m$, reflecting the high genetic diversity within this region (*Figure 3b*). Tajima's D was significantly higher throughout the ZAL2$^m$ outlier region compared with the genomic background (p<0.001, permutation test) (*Figure 3a*).

The increase of nucleotide diversity, high Tajima's D, and long branch length all suggest that balancing selection has impacted the ZAL2$^m$ outlier region. Concordantly, we identified a signature of balancing selection in the ZAL2$^m$ outlier region on the basis of allele frequency spectrum ($\beta$ statistics) computed using BetaScan, which detects signatures of balancing selection by looking for an excess of genomically proximate SNPs with a similar allele frequency (*Siewert and Voight, 2020*). Visual examinations of the ZAL2$^m$ outlier region also revealed the presence of potential long-range haplotypes that likely reflect long-term balancing selection and several recombination events within the haplotypes (*Figure 3c*). Notably, we identified two major putative haplotypes, which we refer to here as 1 and 2, consisting of 17 and 30 ZAL2$^m$ chromosomes, respectively. A possible third haplogroup was also identified, although the sample size (n=2) was too small to warrant further analyses. The genetic differentiation between the two ZAL2$^m$ outlier haplogroups was lower than the differentiation between either haplogroup and ZAL2 (*Figure 3d*), indicating that the haplogroups

**Table 2.** List of protein-coding genes inside the ZAL2m outlier region.

| Gene | Scaffold | Start | End | π ZAL2 | π ZAL2$^m$ | TaD ZAL2 | TaD ZAL2$^m$ | $D_{XY}$ |
|------|----------|-------|-----|--------|-----------|----------|-------------|----------|
| KCNS3 | NW_005081621.1 | 97089 | 110512 | 2.75E-04 | 8.43E-04 | −1.2953 | −0.8658 | 0.011281 |
| MSGN1 | NW_005081621.1 | 160375 | 160897 | NA | 6.10E-04 | NA | −0.1138 | 0.003056 |
| GEN1 | NW_005081621.1 | 175791 | 198245 | 3.52E-04 | 2.22E-03 | −1.1302 | 1.1728 | 0.011647 |
| SMC6 | NW_005081621.1 | 198452 | 244136 | 3.94E-04 | 2.27E-03 | −1.0088 | 0.8548 | 0.011901 |
| MYCN | NW_005081621.1 | 1179492 | 1184761 | 2.59E-04 | 6.18E-04 | −0.5898 | 0.2116 | 0.005173 |
| DDX1 | NW_005081621.1 | 1432697 | 1452535 | 3.41E-04 | 2.42E-03 | −1.5819 | 0.8752 | 0.014699 |
| NBAS | NW_005081621.1 | 1454601 | 1615580 | 2.93E-04 | 2.17E-03 | −1.6271 | 2.2291 | 0.012296 |
| TRIB2 | NW_005081621.1 | 2596178 | 2616950 | 3.45E-04 | 1.89E-04 | −1.0009 | −0.8376 | 0.011498 |
| LPIN1 | NW_005081621.1 | 3012153 | 3061203 | 2.73E-04 | 3.27E-04 | −1.7799 | −0.7252 | 0.015295 |
| GREB1 | NW_005081621.1 | 3100814 | 3165186 | 2.25E-04 | 1.72E-04 | −1.8299 | −0.7351 | 0.01458 |
| E2F6 | NW_005081582.1 | 24475 | 46577 | 1.90E-04 | 1.59E-04 | −1.5978 | −1.0219 | 1.4E-02 |
| ROCK2 | NW_005081582.1 | 50993 | 155170 | 2.62E-04 | 6.34E-04 | −1.46 | −0.2362 | 1.4E-02 |
| KCNF1 | NW_005081582.1 | 331424 | 333991 | 9.85E-05 | 2.41E-04 | −0.2519 | −0.8641 | 5.9E-03 |
| PDIA6 | NW_005081582.1 | 415853 | 431919 | 3.74E-04 | 1.49E-04 | −1.1579 | −1.2521 | 1.2E-02 |
| ATP6V1C2 | NW_005081582.1 | 431766 | 454886 | 2.56E-04 | 6.94E-04 | −1.7535 | −0.2391 | 1.4E-02 |

have evolved since the split between ZAL2 and ZAL2$^m$. This interpretation is also consistent with the inferred phylogeny (*Figure 3b*). Note that the sequencing depth for the two haplogroups did not differ significantly (p=0.78, *Figure 3—figure supplement 3*). The ZAL2$^m$ outlier regions exhibited an excess of non-synonymous SNPs compared with synonymous SNPs (*Figure 2d*), as well as an excess of intermediate frequency minor alleles (*Figure 3—figure supplement 4*). Neither geographic location of collection nor sex was associated with any distinct patterns by haplogroups in a PCA analysis using genetic variants in the region (*Figure 3—figure supplement 5*; although note the breeding location of the birds from two of the locations was unknown). Together, these results solidly implicate balancing selection in the evolution of the ZAL2$^m$ outlier region. In other words, multiple haplotypes of the ZAL2$^m$ chromosome are maintained within the population of white-striped birds via balancing selection.

We looked for phenotypic consequences of the putative ZAL2$^m$ haplogroups using the subset of white-striped birds from the sequencing samples for which we had phenotypic data (n=6 for haplogroup 1, n=12 for haplogroup 2, see *Zinzow-Kramer et al., 2015*; *Horton et al., 2014* for details). White-striped birds with the H1 vs the H2 versions did not differ significantly in aggressive behavior (measured during simulated territorial intrusions), gonad size, tarsus length, cloacal protuberance volume, or any other measure when using the Benjamini-Hochberg correction for multiple testing to avoid false positives (more than 50 tests were performed).

We next tested for effects of haplogroup on the expression of genes inside the ZAL2$^m$ outlier region. In the subset of white-striped birds for which we had haplogroup information, we examined RNAseq data from two brain regions, the hypothalamus (Hyp) and the ventromedial arcopallium (AMV, the functional homolog of the medial amygdala, formally named the nucleus taeniae of the amygdala) (see Materials and methods and *Supplementary file 4* for details). We found that the gene *GREB1* (growth regulating estrogen receptor binding 1) was more highly expressed in haplogroup 2 (unadjusted p=0.0195) in AMV. Additionally, there was a trend for the gene *KCNS3* to be more highly expressed in both Hyp (unadjusted p=0.0511) and AMV (unadjusted $p$ = 0.1567). This gene encodes a voltage-gated channel subunit that in humans and mice is specific to fast-spiking parvalbumin (inhibitory) neurons (*Georgiev et al., 2014*; *Miyamae et al., 2021*). Although no genes were significantly differentially expressed at the genome-wide level, our current findings provide potentially interesting candidates to explore further in the context of behavioral differences between the morphs.

## Candidates for positive selection in ZAL2 and ZAL2$^m$

Our work has provided a refined map of genetic differentiation between the ZAL2 and ZAL2$^m$ chromosomes (*Figure 4—figure supplement 1*). Interestingly, three scaffolds near the chromosomal end (i.e. at the beginning of the p-arm, as well as at the end of the q-arm in the genomic coordinate of the ZAL2 chromosome), exhibited both decreased $F_{ST}$ values and a reduced rate of fixed genetic differences (D$_f$). One possible explanation for these findings is that these regions represent a younger evolutionary stratum, which is possible because ZAL2$^m$ contains a series of nested inversions that occurred at different times (*Lahn and Page, 1999*; *Xu et al., 2019*; *Bergero et al., 2007*). Another possibility is that the inflated within-ZAL2 nucleotide diversity (see *Figure 3a*) within the sampled population reduced the $F_{ST}$ estimate.

By leveraging our resource of a haplotype map of both the ZAL2 and ZAL2$^m$ chromosomes, we investigated signatures of positive selection in each chromosome. We identified 216 20 kB regions on ZAL2 exhibiting a significantly elevated H-statistic (empirical p-values <0.05), which is a measure of the average length of the stretches of linkage disequilibrium between the haplotypes (*Schlamp et al., 2016*; *Messer Lab — Resources, 2014*, Materials and methods). One notable stretch, which included four top candidate 20 kB regions (Scaffold NW_005081582.1, 480–520 kb and 920–960 kb) showed evidence of positive selection on both the ZAL2 and ZAL2$^m$ chromosomes (blue shaded area in *Figure 4—figure supplement 2*). On the basis of previous literature (*Thomas et al., 2008*), this region on ZAL2$^m$ may be placed on the end of that chromosome. The elevated H-statistic in this region cannot be explained by the recombination rate, as recombination is low across the entire length of ZAL2$^m$ (*Figure 4—figure supplement 3*). On the ZAL2 chromosome, in contrast, recombination on the chromosome ends is increased (*Figure 4—figure supplement 3*), as has been reported in other species (*Nachman and Churchill, 1996*; *Barton et al., 2008*). We did not, however, observe an elevated H-statistic in these regions (*Figure 4—figure supplement 2a*). Note that a structurally

resolved chromosome-level assembly does not yet exist for the white-throated sparrow. Thus, the precise location of these regions on the ZAL2$^m$ chromosome still need confirmation.

We also identified a long stretch (~6 Mbp) showing an overall elevated H-statistic in another region of ZAL2 (red shaded area in *Figure 4—figure supplement 3a*, *Supplementary file 5*). This region exhibited the lowest estimates of nucleotide diversity within ZAL2 and the highest estimates of $F_{ST}$ between the two chromosome types. Overall, there were 68 genes located in regions with a significant (p<0.05) H-statistic for ZAL2$^m$ and 109 genes in regions with a significant H-statistic for ZAL2 (*Supplementary file 5*), meaning that these genes are inside regions showing evidence of a selective sweep. These observations suggest that selective sweeps took place on both ZAL2 and ZAL2$^m$ chromosomes after the rearrangement events.

## Antagonistic selection influences ZAL2$^m$ gene expression

Some of the behaviors that differ between morphs, namely aggression, are predicted by the expression of several genes located inside the ZAL2/2$^m$ rearrangement, such as *ESR1* and *VIP* (*Merritt et al., 2020*; *Maney et al., 2020*; *Horton et al., 2020*; *Zinzow-Kramer et al., 2015*; *Horton et al., 2014*). We hypothesize that the ZAL2$^m$ supergene region is enriched for alleles that have been shaped by antagonistic selection. Specifically, we predict alleles within the ZAL2$^m$ supergene region that are beneficial for birds of the white-striped morph, but disadvantageous for birds of the tan-striped morph. Similarly, we predict that the ZAL2 chromosome is enriched for genes with alleles that benefit birds of the tan-striped morph (*Maney et al., 2020*).

To test this hypothesis, we examined whether allelic bias in the expression of genes inside the rearrangement was associated with differential expression of the same genes between the tan- and white-striped morphs (lists of genes showing differential expression or allelic bias can be found in *Supplementary file 6*). Here, allelic bias is defined as the differential expression of the ZAL2 *versus* ZAL2$^m$ alleles in white-striped birds. Allelic bias in expression would indicate a role of *cis*-regulatory variants—i.e., variants in non-coding promotor, enhancer, or silencer regions regulating a gene—in functionally altering the expression of that gene. For these tests, we used data on gene expression in the brain only, as heart and liver tissue samples were limited to white-striped birds.

First, we found that among the genes that were differentially expressed between morphs, more of these genes are located inside the rearranged region on ZAL2/2$^m$ than expected by chance, pointing toward *cis*- (as opposed to *trans*-) regulatory differences underlying morph differences in expression ($X^2$ tests showed highly significant effects for both brain regions in adults and nestlings of both sexes, with false discovery rate, FDR, correction to account for the multiple statistical comparisons performed using the Benjamini-Hochberg correction) (*Figure 4a*). To explore this further, we next tested whether the differentially expressed genes showed greater allelic bias than genes without differential expression. Consistent with what we previously reported using only males (*Sun et al., 2018*), we found that differential expression significantly predicted the degree of allelic bias in expression of that gene ($X^2(2)=664.16$, $p<2.2 \times 10^{-16}$, controlling for sequencing batch and brain region, see Materials and methods). In addition, tan-biased genes showed greater ZAL2 allelic bias and white-biased genes showed greater ZAL2$^m$ allelic bias than did genes that were not differentially expressed between morphs (post-hoc T>W vs T=W: $z=-16.87$; post-hoc W>T vs T=W: $z=19.63$, $p<2.2 \times 10^{-16}$; post-hoc W>T vs T>W: $z=26.35$, $p<2.2 \times 10^{-16}$) (*Figure 4b*). These findings suggest that gene expression differences between the morphs are driven, in part, by evolutionary changes in the regulatory regions of genes captured by the rearrangement.

We also looked for a relationship between gene expression and signatures of positive selection on both the ZAL2 and ZAL2$^m$ chromosomes. We predicted that positive selection on *cis*-regulatory alleles on ZAL2$^m$ would, on average, increase the expression of the ZAL2$^m$ alleles and that positive selection on ZAL2 would increase the expression of ZAL2. We therefore asked whether genes located in regions with evidence for a selective sweep on either ZAL2 or ZAL2$^m$ (as indicated by the H-statistics for 50 kb windows) were also likely to exhibit bias in gene expression in the brain. For the ZAL2$^m$ chromosome, we found that the ZAL2$^m$ H-statistic was a significant, positive predictor of bias in brain gene expression toward the ZAL2$^m$ allele, controlling for sequencing batch and brain region ($X^2(2)=40.17$, $p=1.893 \times 10^{-9}$; $t=5.114$, $p=3.24 \times 10^{-7}$) (see Materials and methods) (*Figure 4c*). Likewise, we found that genes that exhibited evidence of positive selection on ZAL2 showed allelic bias toward the ZAL2 allele ($X^2(2)=24.231$, $p=5.475 \times 10^{-6}$; $t=-3.151$, $p=0.00164$) (*Figure 4d*). Taken together, these results

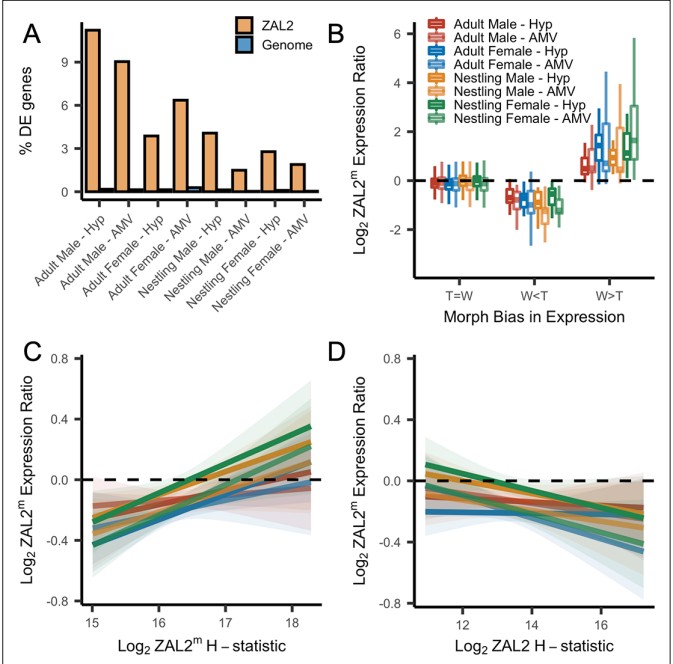

**Figure 4.** Evidence for antagonistic selection driving ZAL2 and ZAL2$^m$ gene expression in the brain. (**A**) shows the percentage of differentially expressed genes that reside inside the rearranged region on ZAL2, vs elsewhere in the genome. The percentage of differentially expressed genes inside vs outside the rearranged region of ZAL2 is higher than expected by chance ($p_{adj}$ <2.2 × 10$^{-16}$ for all comparisons). (**B**) shows log$_2$ ZAL2$^m$ expression ratios for genes that were more highly expressed in white-striped birds (W>T), genes more highly expressed in tan-striped birds (T>W) and those that that do not significantly differ between morphs (T=W). (**C**) Log$_2$ ZAL2$^m$ expression ratios are plotted vs the Log$_2$ ZAL2$^m$ H-statistic for each category of sample. Hypothalamus (Hyp), Ventromedial arcopallium (AMV). (**D**) Log$_2$ ZAL2$^m$ expression ratio are plotted vs the Log$_2$ ZAL2 H-statistic.

The online version of this article includes the following source data and figure supplement(s) for figure 4:

**Source data 1.** Percent of Differentially Expressed genes on ZAL2 vs rest of genome.

**Source data 2.** RNAseq allele specific expression data for brain in long format merged with morph bias and H-scan values.

**Figure supplement 1.** Genetic differentiation between ZAL2 and ZAL2$^m$ is reduced at the ends of the chromosomal arms.

**Figure supplement 2.** Both ZAL2 and ZAL2$^m$ have experienced selective sweeps.

**Figure supplement 3.** Population recombination rates in ZAL2 are likely higher in the chromosome end.

suggest that gene expression on ZAL2 and ZAL2$^m$ is driven by selection for *cis*-regulatory alleles that benefit each morph.

## Candidate genes of interest

Our analyses uncovered many candidate genes that may contribute to the phenotypic differences between the tan- and white-striped morphs. To identify candidates, we performed gene ontology analysis on the set of genes that were either (1) within a region that exhibited evidence of positive selection on either ZAL2 (n=109 genes) or ZAL2$^m$ (n=68 genes) (See *Supplementary file 7*), (2) exhibited allelic bias in six or more tissues (of the eight total brain tissue analyses) (n=249 genes), or (3) exhibited consistent allelic bias across all samples from either one brain region, sex, or age group (i.e. tissue-, sex-, or age-specific expression) (See *Supplementary file 7*, n=117 genes). The rationale for the inclusion of this final group is that genes that show a tissue-specific pattern of allelic bias are strong candidates for genes with adaptive significance, as changes in gene regulation have long been shown to play a role in adaptive evolution (*Wray, 2007*).

Using this set of 409 unique genes, we found the strongest enrichment for genes in the categories 'behavior' ($p_{FDR}$ = 0.0252) and 'metabolic process' ($p_{FDR}$ = 0.0252). The behavior category included

genes such as *GRIK2* (glutamate receptor ionotropic, kainate 2), *GRM1* (metabotropic glutamate receptor 1), *NRXN1* (Neurexin-1), *OPRM1* (μ-opioid receptor), *HTR1B* (5-hydroxytryptamine receptor 1B, i.e. serotonin receptor), *MEIS1* (Meis homeobox 1), and *PAK5* (serine/threonine-protein kinase PAK 5). Additionally, there was enrichment for several gene ontology categories related to metabolism, with 205/409 (50.1%) of these candidate genes involved in the broad category 'metabolic processes' ($p_{FDR}$ = 0.0210). We also found significant enrichment in the category 'regulation of multicellular organismal process' ($p_{FDR}$ = 0.0342). This category included several additional interesting candidate genes, including *ESR1* (estrogen receptor alpha), *NR2E1* (nuclear receptor subfamily 2 group E member 1), *HCRTR2* (orexin receptor type 2), *SYNDIG1* (synapse differentiation-inducing gene 1), *DISC1* (disrupted in schizophrenia 1), and *ESRRG* (estrogen-related receptor gamma).

Allelic bias toward the ZAL2 allele could result from decreased expression of the ZAL2$^m$ allele due to degeneration. Allelic bias in the other direction, toward ZAL2$^m$, is more consistent with adaptation (*Martinez-Ruiz et al., 2020*). We therefore examined relationships between allelic bias and positive selection. We found that 97/359 (27.8%) genes exhibiting a signal of allelic bias showed a bias toward the ZAL2$^m$ allele. Four of those 97 genes also overlapped regions exhibiting a significant H-statistic on ZAL2$^m$: *PDSS2* (all trans-polyprenyl-diphosphate synthase), *XRN2* (5'–3' exoribonuclease 2), *RPS27A* (Ubiquitin-40S ribosomal protein S27a), and *TCTE1* (T-complex-associated-testis-expressed 1).This result implicates these four genes in adaptive ZAL2$^m$ phenotypes, providing novel candidate genes for further exploration.

Looking at our results from a different perspective, of the 409 genes that exhibited evidence either of allelic bias or positive selection (by the H-statistic), 55 exhibited evidences of both. These genes are especially interesting candidates for mediating phenotypic differences between the morphs. Three of these genes were located in areas of elevated H-statistic on both the ZAL2 and ZAL2$^m$ chromosomes: *PDSS2*, *ATP6V1C2* (ATPase H+Transporting V1 Subunit C2), and *GHRL1* (Grainyhead Like Transcription Factor 1). Both *ATP6V1C2* and *GHRL1* were in the shared region highlighted in blue in *Figure 4—figure supplement 2*, suggesting that this region may be an especially important evolutionary hotspot in the divergence between the morphs. Overall, our analysis revealed a number of candidate genes for which positive selection appears to have affected the regulation of genes on both chromosomes.

## Discussion

Understanding evolutionary processes that contribute to the origin and maintenance of supergenes can elucidate the links between genes and complex phenotypes such as behavior. One of the most critical processes in the evolution of supergenes is the suppression of recombination (*Charlesworth, 2016*), a consequence of which is genetic degeneration. Such processes have been extensively studied in the case of non-recombining sex chromosomes (*Barton and Charlesworth, 1998*; *Charlesworth, 2012*; *Charlesworth and Charlesworth, 2000*), but their role in the early evolution of non-recombining autosomes is not well understood. The ZAL2$^m$ supergene captures a snapshot of an early stage of supergene evolution. We and others have previously observed weak signatures of degeneration on the ZAL2$^m$ chromosome (*Tuttle et al., 2016*; *Sun et al., 2018*). In this study, using a large, newly generated genomic sequencing and population genomic data set, we demonstrate pervasive signatures of genetic degeneration at multiple levels, from genomic contigs to SNP density. Regardless, the degeneration is weak at most, and it appears that recombination has not entirely ceased on ZAL2$^m$. It is likely that this low level of recombination occurs in rare ZAL2$^m$ homozygotes (*Horton et al., 2013*). Therefore, the case of the ZAL2$^m$ supergene illustrates that complex phenotypes including alternative plumage morphs and life history strategies, can evolve prior to the cessation of recombination and despite substantial genetic degeneration.

As is evident from research on non-recombining sex chromosomes, both positive and antagonistic selection can further the differentiation between those chromosomes (*Bachtrog, 2004*; *Bachtrog and Charlesworth, 2002*; *Singh et al., 2014*; *Rice, 1984*; *Vicoso and Charlesworth, 2006*; *Mank, 2012*). Previous studies of the ZAL2$^m$ supergene revealed no or only weak evidence of positive selection at the molecular level (*Tuttle et al., 2016*; *Sun et al., 2018*), despite evidence that allelic biases in expression may contribute causally to phenotypic differences between white- and tan-striped morphs (*Merritt et al., 2020*). Here, using extensive molecular data, we detected a strong signature of antagonistic selection based on gene expression profiles and identified several regions of positive selection on both ZAL2 and ZAL2$^m$. We also found that allele-specific gene expression in the brain

was associated with these signatures of positive selection on both the ZAL2 and ZAL2ᵐ chromosomes, suggesting that this selection has acted on regulatory regions of genes. The finding that genes within non-recombining chromosomes experience either positive or antagonistic selection has been reported for *Drosophila* sex chromosomes, as well as for the social supergene of fire ants (*Solenopsis invicta*) (**Martinez-Ruiz et al., 2020**; **Chang et al., 2022**; **Pracana et al., 2017**). This phenomenon is not universal, however, signatures of antagonistic selection were not found in the mating-type chromosomes of the anther-smut fungus (*Microbotryum lychnidis-dioicae*) (**Bazzicalupo et al., 2019**).

Our analyses identified several candidate genes that merit further exploration. For example, several genes that in other species interact with the actions of estrogen receptor alpha (ERα), an important player in the behavioral differences between white-throated sparrow morphs (**Merritt et al., 2020**), also emerged as candidate genes in our analyses. These genes included *NCOA7* (Nuclear Receptor Coactivator 7), a transcriptional coactivator of *ESR1*, and *ESRRG*, a nuclear receptor that interacts with estrogen-responsive elements in the genome (**Festuccia et al., 2018**; **Lazennec et al., 1997**). *NR2E1*, a recently de-orphanized nuclear receptor that binds oleic acid to regulate neurogenesis (**Kandel et al., 2022**), is another exciting candidate gene. Additionally, we identified several genes–*PDSS2*, *XRN2*, *RPS27A*, *TCTE1*, *ATP6V1C2*, and *GRHL1*–that have not previously been linked to the morph differences, but convergent lines of evidence suggest that they may play a role in the evolution of the ZAL2ᵐ supergene. *GHRL1*, for example, is a transcription factor necessary for the development of epithelial tissues and is enriched in the neonatal ventromedial hypothalamus of mice (**Kurrasch et al., 2007**). Its function in the brain, however, remains unknown. Interestingly, the ZAL2ᵐ breakpoint has been mapped to ~12 kB downstream of *GRHL1* (*40*), perhaps accounting for the observed patterns of divergence.

Indeed, we found evidence of selective sweeps in several regions near the predicted breakpoint on ZAL2ᵐ. Positive selection near inversion breakpoints has been found in other species including *Drosophila* (**Evans et al., 2007**). Furthermore, we observed signatures of positive selection on large regions of ZAL2, which was reminiscent of the finding that the X chromosome of primates has been targeted by strong positive selection (**Nam et al., 2015**). This positive selection on ZAL2 could indicate that recessive alleles on that chromosome experience increased selection in birds of the white-striped morph, and are thus in direct conflict with genes on ZAL2ᵐ. It is worth noting, however, that allelic bias is not necessarily indicative of selection; bias in expression toward the tan-allele could be caused by degeneration on ZAL2ᵐ, for example. Bias toward the ZAL2ᵐ allele could be caused by a transposable element in the promoter region, driving aberrant expression. Convergent lines of evidence point toward a role for positive selection throughout the ZAL2ᵐ supergene variant, but the mechanisms underlying allelic bias may vary for each individual gene.

Remarkably, we also discovered clear signatures of balancing selection maintaining two long-range haplotypes within the ZAL2ᵐ supergene. We were surprised by this finding, as similar long-range haplotypes have not, to our knowledge, been previously reported for non-recombining chromosomal polymorphisms. It has been speculated that balancing selection is less common in sexually reproducing species than in self-fertilizing species (**Glémin, 2021**), in which the effective recombination rate is lowered due to the extreme degree of inbreeding. Our results suggest that balancing selection may function to maintain genetic variability in the face of reduced recombination, and we might expect to see other instances of balancing selection within supergenes.

The phenotypic effects of these balanced haplotypes remain unknown. Two out of the fifteen genes inside the balanced ZAL2ᵐ outlier region showed signatures of differential gene expression between the haplogroups. One of these genes is *GREB1*, which in mammals is a downstream target of ERα (**Gegenhuber et al., 2022**). This finding is notable because, in white-throated sparrows, ERα is causally related to the behavioral differences between morphs (**Merritt et al., 2020**; **Horton et al., 2014**). Thus, the ZAL2ᵐ haplogroup may interact in important ways with ERα-mediated phenotypes in white-striped birds, although further research is needed to test this hypothesis.

Taken together, our results demonstrate that the ZAL2ᵐ supergene is far from a degenerating counterpart of a 'normal' autosome. Rather, it is an active arena of multiple evolutionary forces. Although most studies of the consequences of suppressed recombination have focused on genetic degeneration, the resulting supergenes also give rise to opportunities for evolutionary innovation (**Charlesworth, 2016**; **Thompson and Jiggins, 2014**; **Schwander et al., 2014**). Our results show that although degeneration is operating on ZAL2ᵐ, dynamic selective forces are occurring simultaneously,

producing divergent complex phenotypes. These results emphasize that supergenes are an important force in adaptive evolution (*Charlesworth, 2016*; *Schwander et al., 2014*; *Wellenreuther and Bernatchez, 2018*) and open the door for future studies to demonstrate and investigate the functional consequences of dynamic natural selection acting inside recombination-suppressed supergenes.

# Materials and methods

## Whole genome sequencing and population genetics

We performed whole genome sequencing of 63 samples (WS: N=49, TS: N=13, and SW: N=1), which included 28 birds captured during the breeding season near Argyle, Maine (ME), USA, 25 birds captured in Atlanta, Georgia (GA), USA during fall migration (November and December), and 10 postmortem samples opportunistically collected from birds who died from building strikes in Chicago, Illinois (IL), USA (see *Balakrishnan et al., 2014*) during the spring or fall migrations. Sequencing reads from these samples, as well as reads from three previously described samples, were mapped to the reference genome assembly (GCF_000385455.1) using Bowtie2 (ver. 2.3.5) with the very-sensitive-local option. Possible PCR duplicates were removed using Picard tools (ver. 2.19). SNP and INDEL calling were conducted using GATK Haplotypecaller (ver. 4.1.2) with the ERC GVCF option and joint genotyping of all samples were performed with Genotype GVCF option. We filtered out SNPs with any missing information, MAF <0.05, meanDP <5, or meanDP >80. Raw sequencing reads are available on NCBI SRA (BioProject PRJNA818012).

## Sequencing and genome assembly of super-white bird

To complement the available short-read sequencing data from the female super-white bird (*Sun et al., 2018*), we generated additional paired-end reads from the same individual (150 ×; insert size of 300 bp and 500 bp) to improve the assembly quality. Genomic DNA was extracted from a 200 mg liver sample from the super-white bird using the Qiagen DNEasy Blood and Tissue DNA kit. Additionally, mate-pair libraries of different insert sizes (insert size of 1 kb, 4–7 kb, 7–10 kb, and 10–15 kb) were prepared and sequenced by the Brigham Young University Genome Sequencing Center. Raw sequencing reads are available on NCBI SRA (BioProject PRJNA818012).

Using these data, we constructed a whole genome de novo assembly of *Z. albicollis*, including the ZAL2$^m$ chromosome. Paired-end sequencing reads were trimmed by Trimmomatic v.0.32 (*Bolger et al., 2014*), and error correction of the trimmed sequencing reads was conducted by Lighter v.1.1.2 (*Song et al., 2014*). Initial contig assembly and scaffolding was conducted by ABySS v.2.1.5 (*Jackman et al., 2017*). We used Gapcloser (*Xie et al., 2014*) to fill the gaps emerging from scaffolding process. The total assembly size was 1058 Mbp (N50 length of 3.1 Mbp), consisting of 160 scaffolds (111.76 Mbp) belonging to the ZAL2$^m$ chromosome with the longest scaffold length of 5.1 Mbp (*Supplementary file 3*).

## Identification of ZAL2/2$^m$ scaffolds

To discriminate scaffolds that originate from the ZAL2 vs the ZAL2$^m$ chromosome, we mapped the super-white assembly against the genome of the House sparrow (*Passer domesticus*), which is the most closely related species with chromosome level assembly using LASTZ v1.03. Scaffolds uniquely mapped to the homologous House sparrow chromosomes with >30% coverage and >85% identity were retained. In the case of multi-mapping scaffolds, we used more stringent criteria with >70% coverage and >85% identity. We conducted the same procedure using the tan-striped reference assembly. The list of matched scaffolds was highly consistent with our previous study, with the addition of several new scaffolds (n=15, sum of scaffold lengths = 129.1 kb). To distinguish sequences inside vs outside of the rearranged (i.e. inverted) regions, we computed the average frequency of heterozygous SNPs in sliding windows of 25 kb size with 1 kb step.

## Genetic differentiation between the ZAL2 and ZAL2$^m$ chromosomes

Utilizing the large amount of newly generated whole genome sequence data, here we identified putatively fixed genetic differences between the ZAL2 and ZAL2$^m$ chromosomes. Briefly, we identified positions at which the genomes of the tan-striped (ZAL2/2) and super-white (ZAL2$^m$/2$^m$) birds are homozygous for different alleles while the white-striped (ZAL2/2$^m$) birds are heterozygous. Note

that the probability of this allelic pattern occurring by random chance is $2.6 \times 10^{-23}$ given the sample size, according to a binomial test. Following this procedure, we obtained a total of 931,424 SNPs and 97,375 insertions and deletions (InDels) between ZAL2 and ZAL2$^m$ chromosomes, increasing the number of putatively fixed SNPs between the two chromosomes beyond previous publications (**Tuttle et al., 2016**; **Sun et al., 2018**). Variant call format files of fixed differences (both SNPs and InDels) are available at DOI: 10.6084 /m9.figshare.19395146. As we expected, a vast majority (N=930,588; 99.91%) of the fixed differences reside in the scaffolds that we previously predicted to be inside the rearrangement (**Sun et al., 2018**). The remaining fixed differences were found in scaffolds that were either too short or that mapped ambiguously to multiple chromosomes. FastEprr was used to estimate the recombination rate for non-overlapping 50 kbp sliding window with default setting (**Gao et al., 2016**).

## Haplotype phasing of whole genome and RNA sequencing data

We performed read-based haplotype phasing of the sequencing data using reads from white-striped individuals (ZAL2/2$^m$ heterozygotes). Briefly, using the 931,424 single nucleotide fixed differences between ZAL2/2$^m$, we assigned sequence reads from the heterozygous birds to the corresponding chromosome of origin (*i.e.* either ZAL2 or ZAL2$^m$) if the paired-end reads were mapped to a region that overlapped at least one fixed difference. Reads from white-striped birds were mapped to *N*-masked (masking the putative fixed differences between ZAL2 and ZAL2$^m$) reference assembly to avoid mapping bias and assigned to their chromosome of origin using SNPsplit v.0.3.2 (**Krueger and Andrews, 2016**). Because the two supergene variants differ in the amount of genetic diversity they carry, and because read mapping algorithms are sensitive to the number of differences between the reference sequence and the read, our approach risks a slight mapping bias. We believe that this slight bias is unlikely to have fundamentally changed our results.

On an average, 8.5% of all reads from white-striped birds were assigned to either the ZAL2 or ZAL2$^m$ chromosomes. In comparison, the size of the chromosomal inversion is estimated to be approximately 9.5% of the total genome (**Thomas et al., 2008**). The ZAL2 and ZAL2$^m$ assigned reads were extracted respectively using Bedtools *bamtofastq*. To call variants for both ZAL2 and ZAL2$^m$ chromosomes, the reads were remapped to the reference genome assembly using Bowtie2 v. 2.3.5. Variant calling was conducted using GATK Haplotypecaller v.4.1.2 with the ERC GVCF option. Vcftools was used to filter out SNPs with any missing information, MAF <0.05, meanDP <5, or meanDP >80. Accession information for all raw RNA sequencing data used is available in **Table 1**.

## Population genomic analysis

Estimates for nucleotide diversity ($\pi$), between population divergence ($d_{XY}$), density of fixed differences ($d_f$), between population difference in allele frequency ($F_{ST}$), and Tajima's D were computed in non-overlapping sliding windows sized 10 kb, 25 kb, and 50 kb, respectively. If a data set has substantial missing points, the estimation of nucleotide diversity is often biased (**Korunes and Samuk, 2021**). Thus, calculation of nucleotide diversity and $d_{XY}$ from the variant call format (VCF) file can be over-estimated since missing sites are not distinguished from invariant (monomorphic) positions in variants-only VCFs. To account for potential inflation of population summary statistics, we considered an average breadth of coverage across samples for each window when we computed nucleotide diversity, $d_{XY}$, and $d_f$. Nucleotide diversity of protein coding sequence was computed using SNPGenie (**Nelson et al., 2015**).

For phylogenetic reconstruction of ZAL2$^m$ outlier windows, we used RaxML v.8.0.2 for 200 kb windows. The targeted window was selected based on the Tajima's D values and haplogroup assignments were determined by manually inspecting genotype plots. For the control region, we concatenated 10 randomly selected regions of 20 kb windows (200 kb). The evolutionary model was set to GTRGAMMA with acquisition bias correction using conditional likelihood method (-m ASC_GTRGAMMA –asc-corr=lewis). The medium ground finch (*Geospiza fortis*) was used as an outgroup species. We conducted bootstrap with 100 replicates.

To examine whether introgression from other species could account for the ZAL2$^m$ outlier window, we computed the D-statistic (**Martin et al., 2015**) using as ingroup genomes the ZAL2 chromosome, the Harris' sparrow chromosome 2, and the ZAL2$^m$ chromosome (P1, P2, and P3, respectively). We

used the medium ground finch as the outgroup species (P4). The D-statistic was computed for four individual white-striped birds with high sequencing coverage.

The analysis of linkage disequilibrium (LD) decay was performed across the entire chromosomal rearrangement. We calculated pairwise $r^2$ values between variant sites using *PopLDdecay* (*Zhang et al., 2019*).

We performed H-scan (*Schlamp et al., 2016*; *Messer Lab — Resources, 2014*) to identify soft and hard sweeps, inferred by extended tracts of homozygosity, using phased haplotypes of ZAL2 and ZAL2$^m$. H-scan outputs were separated into 25 kb non-overlapping windows. As a representative summary statistic, we chose the maximum H-statistic value for each window. To calculate the empirical p-value, we first binned genomic windows into 50 SNP increments based on the number of SNPs. When a window included more than 100 SNPs, all the windows were merged into one bin. On the basis of the ranking of summary statistics in each bin, we calculated an empirical p-value for each window. Candidate regions of positive selection were defined as those with an empirical p-value less than 0.05.

## Gene expression analyses

We examined both gene expression divergence and allele-specific expression using three RNAseq data sets from white-throated sparrows (four types of tissue from four separate samples of birds, see *Table 1*). The first data set consisted of gene expression in male white-throated sparrows collected during the breeding season. In those birds, the samples comprised two brain regions, the Hyp and the AMV (*Sun et al., 2018*; *Zinzow-Kramer et al., 2015*), a functional homolog of the medial amygdala (*Mello et al., 2019*). The second data set consisted of gene expression in the same two brain regions (Hyp and AMV) in adult females collected during the breeding season as well as male and female nestlings (see *Sun et al., 2021* for details). The third data set consisted of gene expression in heart and liver tissue in white-striped males collected during migration, then housed in captivity under two lighting conditions (long days and short days) and sampled at two time points during the day (ZT6 and ZT18) (*Horton et al., 2019*). In statistical analyses, the females and nestlings of each sex were treated as separate 'batches' of RNAseq such that there were five total batches: brain samples from adult males, adult females, nestling males, nestling females, and liver/heart samples from adult males. Each tissue was nested within batch to account for repeated sampling from the same individuals.

RNAseq reads were aligned to a reference genome *N*-masked for putative fixed differences for both chromosomal polymorphisms present in this species (i.e. ZAL2/2, ZAL3$^a$/3$^a$,) using *HiSat2* (*Kim et al., 2015*). To examine allele-specific expression, we used *SNPsplit* v.0.3.2 to assign mapped reads to ZAL2 or ZAL2$^m$ for the white-striped samples. Transcripts were quantified using *StringTie* (*Pertea et al., 2016*) and differential expression and allelic-specific expression analyses were performed using *DESeq2* (*Love et al., 2014*). To test for differential expression between the morphs, we used the following model in *DESeq2*: 'design = ~morph'. To test for allelic bias (AB), we used the size factors generated for each sample in the previous step, and then used the following model: 'design = ~individual + allele'. To perform hypothesis testing, we used linear mixed models with tissue nested in sequencing batch as random effects using the R package *lme4*. To test for the effect of interest, we then performed a likelihood ratio test to compare a full model to a reduced model with the factor of interest removed using the anova function. Where applicable, we used the summary function to perform post-hoc comparisons between multiple groups.

To test for significant allelic bias in expression within each sample type, we used a Wilcoxon rank sum test to test whether the ratio of ZAL2$^m$ to ZAL2 expression differed significantly from 0.5, applying an FDR correction for multiple testing. To test for an association between allelic bias and the number of mutations within a gene, we computed the per-base number of mutations by dividing the number of mutations by the gene length and then sorted the genes into deciles. We also computed the number of fixed mutations within 1 kB upstream of the transcription start site (TSS). We then tested whether there was an association between allelic bias and the decile rank or number of mutations within 1 kB upstream of the TSS using the following linear model: Log2 AB ~Variable + (1|Batch:-Tissue). To test for overrepresentation of morph-biased genes located inside the rearrangement, we performed a two-sample test for equality of proportions for each brain sample type only. For this test, we compared the proportion of differentially expressed genes, out of the 1007 genes inside the rearranged region on ZAL2, to the proportion out of the 13,369 genes located elsewhere in the genome.

For this comparison, we used the *prop.test* function in R to perform a two-proportions Z-test, applying an FDR correction for multiple testing. To test whether the morph bias in expression of a gene significantly predicted allelic bias for that gene, we grouped genes into three categories: those that were more highly expressed in the white-striped morph (W>T), those more highly expressed in the tan-striped morph (T>W) and those that that did not differ significantly between morphs (T=W). Using the allelic bias for each gene, we then tested whether allelic bias differed among these categories using the following linear mixed model: $\log_2$ AB ~MorphBias_Category + (1|Batch:Tissue), followed by Bonferroni-corrected pairwise post-hoc tests using the 'ghlt' function in *multcomp* package. Note that heart and liver tissue samples were obtained from white-striped birds; because we did not have data on differential expression by morph for these tissues, they were excluded from this analysis, as well as the tests for overrepresentation described above.

To test whether haplogroup affects gene expression, we merged the gene count matrices for the male (H1: n=7, H2: n=3) and female (H1: n=3, H2: n=3) white-striped birds that were also included in the whole genome sequencing dataset and tested for an effect of haplogroup using the following model: design ~sex + haplogroup. Only one gene in Hyp (Geranylgeranyl Diphosphate Synthase 1, *GGPS1*) and one gene in AMV (RUN And FYVE Domain Containing 4, *RUFY4*) were differentially expressed at the genome-wide level. Neither of these genes are located in the $ZAL2^m$ outlier region and only *GGPS1* is located on ZAL2. Thus, we report only unadjusted p-values for genes inside the $ZAL2^m$ outlier region that were differentially expressed (both unadjusted and adjusted p-values for all genes in the ZAL2m outlier region are reported in *Supplementary file 4*).

To examine the relationship between the H-statistic and allelic bias in expression, we computed the average H-statistic for each gene. Each gene was assigned the H-statistic value of the 20 kb bin overlapping the gene (or the average of multiple 20 kb bins, if a gene overlapped two bins). We then used a multiple linear regression model to examine the relationship between the $\log_2$ASE and the $\log_2$H-statistic of a gene for both ZAL2 and $ZAL2^m$ separately, using the following model: Log2 AB ~ $\log_2$H-statistic + (1|Batch:Tissue). Additionally, we performed functional enrichment analyses for the genes in regions with a significant H-statistic using ToppFun with human homolog HGNC gene names (*Chen et al., 2009*).

## Acknowledgements

We thank the members of the Maney lab who performed field work and collected the samples from GA and ME, as well as provided helpful feedback throughout the preparation of this manuscript. K.E. Grogan performed the RNA-sequencing for the samples from adult females and nestlings. David Willard (Collection Manager—Birds, Field Museum of Natural History, Chicago, IL) collected and provided access to the samples from IL. CNB thanks Dr. Elaina Tuttle for passing these samples on to him, and for introducing him to the white-throated sparrow research. This work was supported by NSF IOS-0723805 to DLM and by NIH 1R01MH082833, NIH R21MH102677, and NSF IOS-1656247 to DLM and SVY.

## Additional information

### Funding

| Funder | Grant reference number | Author |
|---|---|---|
| National Science Foundation | IOS-0723805 | Donna L Maney |
| National Institutes of Health | 1R01MH082833 | Donna L Maney<br>Soojin V Yi |
| National Institutes of Health | R21MH102677 | Donna L Maney<br>Soojin V Yi |
| National Science Foundation | IOS-1656247 | Donna L Maney<br>Soojin V Yi |

| Funder | Grant reference number | Author |
|--------|------------------------|--------|

The funders had no role in study design, data collection and interpretation, or the decision to submit the work for publication.

## Author contributions

Hyeonsoo Jeong, Data curation, Formal analysis, Validation, Investigation, Visualization, Methodology, Writing – original draft, Writing – review and editing; Nicole M Baran, Formal analysis, Investigation, Visualization, Methodology, Writing – original draft, Writing – review and editing; Dan Sun, Conceptualization, Data curation, Formal analysis, Investigation, Methodology, Writing – review and editing; Paramita Chatterjee, Investigation, Methodology, Sample collection, generating sequencing libraries, and performing sequencing; Thomas S Layman, Investigation, Methodology, Sample collection, generating sequencing libraries, and performing sequencing; Christopher N Balakrishnan, Conceptualization, Investigation, Methodology, Writing – review and editing, Sample acquisition; Donna L Maney, Soojin V Yi, Conceptualization, Supervision, Funding acquisition, Investigation, Methodology, Writing – original draft, Project administration, Writing – review and editing

## Author ORCIDs

Hyeonsoo Jeong ⬦ http://orcid.org/0000-0002-5565-2685
Nicole M Baran ⬦ http://orcid.org/0000-0001-6270-0625
Christopher N Balakrishnan ⬦ http://orcid.org/0000-0002-0788-0659
Donna L Maney ⬦ http://orcid.org/0000-0002-1006-2358
Soojin V Yi ⬦ http://orcid.org/0000-0003-1497-1871

## Ethics

All procedures involving live animals were approved by the Emory University Institutional Animal Care and Use Committee (IACUC Protocols PROTO201800016) and with appropriate state and federal (Maine Inland Fisheries and Wildlife, Federal U.S. Geological Survey, and U.S. Fish and Wildlife Service) permits. All procedures, including banding, blood sampling, behavioral studies, capture, and euthanasia, adhered to the Guidelines to the Use of Wild Birds in Research.

## Decision letter and Author response

Decision letter https://doi.org/10.7554/eLife.79387.sa1
Author response https://doi.org/10.7554/eLife.79387.sa2

# Additional files

## Supplementary files

- Supplementary file 1. Sequencing sample information.
- Supplementary file 2. Population genetics sequencing information.
- Supplementary file 3. Genome assembly summary statistics.
- Supplementary file 4. ZAL2$^m$ outlier region differential expression analysis.
- Supplementary file 5. H-scan significant gene list.
- Supplementary file 6. Significant differential expression and allelic bias gene lists.
- Supplementary file 7. Candidate gene lists.
- MDAR checklist

## Data availability

Whole genome sequencing data have been deposited in NCBI BioProject under accession code PRJNA818012. RNA sequencing data from adult females and nestling brain tissue have been deposited in NCBI BioProject under accession code PRJNA657006. Previously published sequencing data also used are listed in Table 1. Variant call format files of fixed differences between ZAL2 and ZAL2$^m$ are available at DOI: https://doi.org/10.6084/m9.figshare.19395146.

The following dataset was generated:

| Author(s) | Year | Dataset title | Dataset URL | Database and Identifier |
|---|---|---|---|---|
| Yi SV | 2022 | Dynamic molecular evolution of a supergene with suppressed recombination in white-throated sparrows | https://www.ncbi.nlm.nih.gov/bioproject/PRJNA818012 | NCBI BioProject, PRJNA818012 |

The following previously published datasets were used:

| Author(s) | Year | Dataset title | Dataset URL | Database and Identifier |
|---|---|---|---|---|
| Maney DL, Horton B, Grogan KE | 2020 | Gene expression differences by morph, sex, and age in adult and nestling white-throated sparrows | https://www.ncbi.nlm.nih.gov/bioproject/PRJNA657006 | NCBI BioProject, PRJNA657006 |
| Zinzow-Kramer WM, Horton BM, McKee CD, Michaud JM, Tharp GK, Thomas JW, Tuttle EM, Yi SV, Maney DL | 2017 | Genes located in a chromosomal inversion are correlated with territorial song in white-throated sparrows | https://www.ncbi.nlm.nih.gov/geo/query/acc.cgi?acc=GSE77186 | NCBI Gene Expression Omnibus, GSE77186 |
| Horton WJ, Jensen M, Sebastian A, Praul CA, Albert I, Bartell PA | 2018 | Transcriptome Analyses of Heart and Liver Reveal Novel Pathways for Regulating Songbird Migration | https://www.ncbi.nlm.nih.gov/geo/query/acc.cgi?acc=GSE116989 | NCBI Gene Expression Omnibus, GSE116989 |

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
