## [Editor Report]

In this important paper, the authors generate and analyze new genome and gene expression data to understand better the evolution of the white-throated sparrow supergene region, which contains 1000 genes and determines whether a bird has a tan or a white stripe. The study convincingly illustrates how the cessation of recombination that results from a chromosomal inversion can become a source of evolutionary novelty. The lack of recombination can result in the accumulation of deleterious variation leading to degeneration, but it can also (as here) facilitate genomic diversification and adaptation. The results will be of interest to a broad array of researchers studying genome architecture and phenotypic diversity and evolution.

---

## [Decision Letter]

**Decision letter after peer review:**

Thank you for submitting your article "Dynamic molecular evolution of a supergene with suppressed recombination in white-throated sparrows" for consideration by *eLife*. Your article has been reviewed by 2 peer reviewers, and the evaluation has been overseen by a Reviewing Editor and George Perry as the Senior Editor. The following individual involved in the review of your submission has agreed to reveal their identity: Yannick Wurm (Reviewer #2).

Essential revisions:

The reviewers are largely positive about your paper, as am I. They have multiple excellent suggestions for your consideration below; please address each comment in your point-by-point response to accompany your revised manuscript. Essential revisions include expanded analytical consideration and discussion concerning the many genes being evaluated here and consideration of the potential role of repetitive element variation in the phenotypic outcomes. I'm personally not one to advocate for the strict use of α=0.05 cutoff for determining statistical 'significance', but I share reviewer #1s view that the breadth of multiple testing corrections should be carefully considered. (And given the breadth of the audience for this paper, you could briefly explain false discovery rate estimation and use it in the main text when it is initially used).

*Reviewer #1 (Recommendations for the authors):*

– Add a few words describing tests and the meaning of the statistics for people less habituated to their use (H-statistics, D-statistic, …). The results of the ABBA-BABA and, particularly, Supplemental Figure 4 are difficult to understand and additional explanations are needed.

Regarding the study of balancing selection:

– lines 311-312. Looking at the figure it would be more reasonable to consider 3 haplogroups within the outlier region (one of them with just very few samples) or even more. The use of just two haplogroups should be justified.

– Is this outlier region heterozygous for all individuals? The text does not offer information about this

– l. 322-327. When looking for phenotype changes associated with the haplogroups, how is it possible to study the phenotypic effect of the haplogroups if they are expected to be in the heterozygous state as a result of balancing selection? Info needed.

– If the test if Supp Figure 7 was not significant after correcting for multiple testing, it does not seem correct to infer that the differences are significant. Remove. (Also, explain FDR testing. I do not know what this means).

– l. 332-338. Corrections for multiple testing, have they been applied? Differences for KCNS3 can not be considered significant for either tissue if p=0.0511 and p = 0.1567 (especially if no correction for multiple testing is applied).

Regarding the study of positive selection:

– l. 345-347. The possible explanations for the reduced differentiation toward the end of the inversion need further explanation

a) Represents a younger evolutionary stratum? If it is part of the same inversion, how would a younger date be possible?

b) How would genetic diversity be increased in this area?

Could it just be the result of an event of recombination in the past, introducing diversity from one lineage into the other? Or the result of the selective sweep (Supp Figure 9)?

– Regions with evidence of positive selection: The text reads (l.352), "four top candidate regions (NW_005081582.1, 480-520kb and 920-960kb)". The legend of the Supplemental Figure 9 reads "A candidate region showing a positive selection on both ZAL2 and ZAL2m (NW_005081582.1, 480-520kb, and 920-960kb)". Is it one region or four regions? What are the terms in parenthesis?

– The text mentions different regions with evidence of selective sweeps, but it is difficult to relate those regions to results: l. 354, peaks in chromosome end in ZAL2; which ones are those?; l. 357 the long stretch of elevated H-statistics in ZAL2, could this be associated with low nucleotide diversity and high Fst? In general, this paragraph is quite confusing.

– I miss some info about the content of the regions with evidence for a selective sweep. It would be nice to know what genes are inside the region where selective sweeps have taken place for both chromosomes. The authors search for ontological or functional enrichment, but it is not clear if differences should be expected at this level and I would like to see info about specific candidate genes that could have been proposed considering previous studies and phenotypic analyses

– Antagonistic selection implies that changes in one chromosome counteract changes in the other one. Not clear how the approach used allows the investigation of the existence of counteracting changes. The text in this section is hard to follow. For example, in line 379 it is difficult to understand what allelic bias means. Or, in line 383, it seems that tan-biased genes can only show greater ZAL2 allelic bias because ZAL2m is not present in those individuals. Please, revise this entire section (starting in l. 364).

Other comments:

– l. 243 Sentence hard to follow "We found evidence that allelic bias in gene expression was associated with the decile rank of the per-base number of non-synonymous fixed differences". Could this be assessed without using deciles: bias in gene expression vs. per-base number of non-synonymous fixed differences.

– l. 288. Please, provide an intuitive explanation of the D-statistic and what it means here. Difficult to see anything in Figure 3a.

– l. 425 Reference 63 corresponds to a web page with software. It is not clear what this reference means here. Please, check references to make sure that there are no errors (for example, see author list in reference 64).

*Reviewer #2 (Recommendations for the authors):*

Line 235: "whether degeneration … resulted in … reduced expression of the ZAL2m allele. " – ZAL2m is a whole supergene (or chromosome) variant. It is not an allele. Probably the following is better: "reduced expression of ZAL2m alleles". Throughout the manuscript, I don't think the authors should say "ZAL2m allele" when referring to the supergene variant. "Alleles carried by the Zal2m supergene variant" could be appropriate.

Figure 4b: How many genes were there with the different types of bias? For genes with a higher expression in Tan than white, the authors suggest that the allelic bias towards the tan allele is a sign of adaptation. But it could simply be that the bias is the result of the ZAL2m allele having degenerated (similar to https://elifesciences.org/articles/55862 ). Conversely, while some higher expression of Zal2m alleles in white birds could be adaptive, some of it can also be that a transposon jumped in front of the promoter and leads to maladaptive high gene expression. We need to be careful with adaptationist interpretations.

Line 354 – finding extended homozygosity runs near the end of the chromosome arm could be expected from lower recombination in this region?

Line 295-298: the authors check for introgression by building a tree from the region of Zal2m that is putatively balancing selection. A large amount of collapsed repetitive DNA in Zal2 can create misleading patterns in phylogenetic reconstruction attempts. This is due to the rapid turnover and jumping of repetitive elements (e.g., what looks like it is here in the reference genome may actually be from somewhere else, or could represent a hybrid genotype of several divergent repeat copies). Focusing on single-copy loci, such as exons of intact single-copy genes, should be preferred (see Stolle Nat Com2022 for one approach).

Examination of allelic expression bias involves mapping to a genome where fixed differences between supergene variants have been masked. This is ok, but because the 2 supergene variants differ in the amount of genetic diversity they carry, and read mapping algorithms are sensitive to the number of differences between the reference sequence and the read, this would lead to a slight mapping bias. That creates a risk here. A more balanced control could have involved masking any genetic variation. However, I think that it is unlikely that doing so would fundamentally change the picture.

Figure 1A shows that ZAL2m has higher overall genetic diversity than ZAL2, yet the text describing genetic diversity also consistently suggests the alternate pattern. For example, Ne derived from synonymous genic diversity is 10x lower in ZAL2m than ZAL2 (from line 222).

Alternatively, the apparent conundrum could come from the use of different filtering criteria and the use of a linear reference sequence where repetitive regions are collapsed. The ZAL2m's genome sequence contains more difficult-to-assemble repetitive DNA, as evidenced by finding smaller scaffolds in Zal2m. This is in line with what is seen in the young fire ant Sb chromosomes (Pracana 2017 Mol Eco finding lower genetic diversity, Stolle 2018 finding more repetitive elements, and more variation in repetitive element content), as well as in some labile young Y chromosomes.

---

## [Author Response]

Essential revisions:The reviewers are largely positive about your paper, as am I. They have multiple excellent suggestions for your consideration below; please address each comment in your point-by-point response to accompany your revised manuscript. Essential revisions include expanded analytical consideration and discussion concerning the many genes being evaluated here and consideration of the potential role of repetitive element variation in the phenotypic outcomes. I'm personally not one to advocate for the strict use of α=0.05 cutoff for determining statistical 'significance', but I share reviewer #1s view that the breadth of multiple testing corrections should be carefully considered. (And given the breadth of the audience for this paper, you could briefly explain false discovery rate estimation and use it in the main text when it is initially used).

Thank you for the careful and thoughtful evaluation of our manuscript. We have added an expanded consideration of candidate genes and their functional implications on lines 450-489 and lines 519-531. Additionally, we have done our best to address concerns related to the role of repetitive element variation, at least where we can address whether our findings are a result of increased structural variation, as opposed to other phenomena. One of the limitations of the current system include the lack of high quality long-range sequence data from the species, thus making it difficult to resolve structural variation and repetitive sequences, so this is not something that we can address in depth in this manuscript. We also have carefully considered our determination of statistical ‘significance’. In one case, a result was removed, but we opted in another instance to keep some borderline findings in the manuscript (see below). We have also made sure to explain false discovery rate estimation, as well as provided other operational definitions for statistics used, in the text.

Reviewer #1 (Recommendations for the authors):– Add a few words describing tests and the meaning of the statistics for people less habituated to their use (H-statistics, D-statistic, …). The results of the ABBA-BABA and, particularly, Supplemental Figure 4 are difficult to understand and additional explanations are needed.

Additional definitions/explanations for the various statistics (H-statistic, D-statistic, & β-statistic) were provided on lines (302-303, D-statistic; 317-318; β-statistic; 365-366, H-statistic). Additionally, in this process and in response to comments from Reviewer #2, we identified and corrected a labeling error that had been made in the supplementary figure (now Figure 3 – supp. 2), as well as several errors in the text. We updated the figure, and made changes to the figure legend, and to the text on lines 301-306 and in the Materials & Methods on line 667-671.

Regarding the study of balancing selection:– lines 311-312. Looking at the figure it would be more reasonable to consider 3 haplogroups within the outlier region (one of them with just very few samples) or even more. The use of just two haplogroups should be justified.

Indeed, there appears an additional haplogroup. However, the sample size of the potential third haplogroup is too small (n=2) to enable in-depth analysis. We thus focused on the two major haplogroups that have moderate numbers of chromosomes. Further sampling would be necessary to pursue deeper investigation. We modified the text on lines 320-323 to clarify this point. We also note that different subregions of the outlier region had different topology when drawing phylogenetic trees, although the presence of two major haplogroups was generally consistent. Once again, we feel this issue can be resolved with larger sample sizes.

– Is this outlier region heterozygous for all individuals? The text does not offer information about this

There seems to be some confusion here, so we have attempted to clarify here and in the text (on lines 332-334). White-striped birds are heterozygous for ZAL2 and ZAL2^m^, so they have only one copy of ZAL2^m^. Our results suggest that within ZAL2^m^, there is a region for which balancing selection has maintained multiple (2, or possibly 3) haplotypes. Each white-striped (heterozygous) bird has only one of the two haplotypes of the ZAL2^m^ chromosome and tan-striped birds are unaffected altogether. This is an example of balanced haplogroups within one of the chromosomes maintained by another instance of balancing selection.

– l. 322-327. When looking for phenotype changes associated with the haplogroups, how is it possible to study the phenotypic effect of the haplogroups if they are expected to be in the heterozygous state as a result of balancing selection? Info needed.

Here we are testing whether white-striped birds with H1 version of ZAL2^m^ differ from whitestriped birds with the H2 version of ZAL2^m^ in any phenotypes. We have attempted to clarify this on lines 337-340.

– If the test if Supp Figure 7 was not significant after correcting for multiple testing, it does not seem correct to infer that the differences are significant. Remove. (Also, explain FDR testing. I do not know what this means).

We have removed this claim regarding tarsus length. We have also explained our correction for multiple testing approach on lines 339-340.

– l. 332-338. Corrections for multiple testing, have they been applied? Differences for KCNS3 can not be considered significant for either tissue if p=0.0511 and p = 0.1567 (especially if no correction for multiple testing is applied).

Here we chose not to apply corrections for multiple testing, and we have attempted to be transparent about this choice. We did not find any significantly differentially expressed at the genome-wide level (and, again, there are substantial limitations to the data we had available for these comparisons), so we opted to share these tentative findings in a transparent way. See lines 350-352.

Regarding the study of positive selection:– l. 345-347. The possible explanations for the reduced differentiation towards the end of the inversion need further explanationa) Represents a younger evolutionary stratum? If it is part of the same inversion, how would a younger date be possible?

It is hypothesized that the origin of the ZAL2^m^ chromosome involves a series of potentially nested inversions (Thomas et al., 2008 Genetics) and now noted on lines 360-361, not unlike what has been observed, for example, for the mammalian Y chromosome. These multiple inversion events could have taken place at different times, which could lead to a younger evolutionary stratum. We cited this possibility due to the strong precedence in other systems. However, we do not have direct evidence that clearly supports this claim.

b) How would genetic diversity be increased in this area?Could it just be the result of an event of recombination in the past, introducing diversity from one lineage into the other? Or the result of the selective sweep (Supp Figure 9)?

We present evidence that ZAL2 genetic diversity is elevated at the chromosome ends in Figure 3a. This could be as a result of increased recombination taking place at the chromosome ends, consistent with what has been reported in the literature (e.g. Nachman & Churchill, 1996, *Genetics*; Barton et al., 2008, *Genetics*). We tested this explicitly and now include Figure 4 – supp. 3, which provides evidence that supports the idea that increased diversity on the ends of the ZAL2 chromosome is a result of increased recombination there (lines 370-376). We do not, however, find evidence of increased selective sweeps taking place on the chromosome ends of the ZAL2 chromosome (see revised Figure 4 – supp. 2a and our expanded description of this result, lines 370-373). ZAL2^m^ shows several broad regions of selective sweeps (for example scaffold NW_005081553.1, the large red colored scaffold on the righthand side of Figure 4 – supp. 2b), but it is worth noting that this area is not in fact the end of the ZAL2^m^ chromosome. Figure 4 – supp. 2 (and all the graphs) shows the genomic coordinates with respect to the *ZAL2* assembly, where the scaffolds have been ordered using BAC sequencing and FISH studies (Thomas et al. 2008). A structurally resolved chromosome-level assembly does not yet exist for the white throated sparrow. Thus, the precise location of these regions on the ZAL2^m^ chromosome are currently unknown, in absence of long-read sequencing and a high quality ZAL2^m^ assembly (now pointed out on lines 373-375). We therefore cannot make strong claims about the exact locations of these regions along the chromosome, unfortunately.

– Regions with evidence of positive selection: The text reads (l.352), "four top candidate regions (NW_005081582.1, 480-520kb and 920-960kb)". The legend of the Supplemental Figure 9 reads "A candidate region showing a positive selection on both ZAL2 and ZAL2m (NW_005081582.1, 480-520kb, and 920-960kb)". Is it one region or four regions? What are the terms in parenthesis?

There are four 20kB regions that are close to each other with significant evidence of positive selection on both the ZAL2 and ZAL2^m^ chromosomes, so part of a single larger region. We have now highlighted this region in blue in Figure 4 – supp. 2. This has also been clarified in the text on lines 367-368 and in the Supplementary figure legend. The terms in parentheses are the scaffold names and genomic coordinates for the reference genome. We have edited the manuscript to clarify these points.

– The text mentions different regions with evidence of selective sweeps, but it is difficult to relate those regions to results: l. 354, peaks in chromosome end in ZAL2; which ones are those?; l. 357 the long stretch of elevated H-statistics in ZAL2, could this be associated with low nucleotide diversity and high Fst? In general, this paragraph is quite confusing.

We have substantially modified this paragraph (lines 463-483) to avoid confusion. We also added shaded blocks to Figure 4 – supp. 2 to aid our presentation. Indeed, regions with elevated H-statistics also exhibit low nucleotide diversity and high Fst, which could have been due to selective sweeps.

– I miss some info about the content of the regions with evidence for a selective sweep. It would be nice to know what genes are inside the region where selective sweeps have taken place for both chromosomes. The authors search for ontological or functional enrichment, but it is not clear if differences should be expected at this level and I would like to see info about specific candidate genes that could have been proposed considering previous studies and phenotypic analyses

The genes in regions that show significantly elevated H-statistics are listed in Supplementary Table 5. Functional genomic resources and tools from this species are still lacking. Thus, we initially hesitated to over-interpret the results, though we believe our results provide an interesting list of candidate genes for future studies to follow up on. In the revised manuscript, we included additional analyses & discussion connecting the positive selection analyses with differential gene expression (in lines 450-489) and provided a list of candidate genes in the new Supp. Table 7.

– Antagonistic selection implies that changes in one chromosome counteract changes in the other one. Not clear how the approach used allows the investigation of the existence of counteracting changes. The text in this section is hard to follow. For example, in line 379 it is difficult to understand what allelic bias means. Or, in line 383, it seems that tan-biased genes can only show greater ZAL2 allelic bias because ZAL2m is not present in those individuals. Please, revise this entire section (starting in l. 364).

We use antagonistic selection to infer selection for divergent alleles on ZAL2 and ZAL2^m^ that benefit individuals of the tan-striped and white-striped morph, respectively. Although allelic bias is an imperfect test of adaptation, we make the assumption that positive selection on *cis*-regulatory alleles on ZAL2^m^ would, on average, increase the expression of the ZAL2^m^ allele and *visa versa*. This framework is similar to what is often used to test for sexually antagonistic selection on the X and Y chromosomes. To test this prediction, for each gene within the rearrangement, we relate differential expression between birds of each morph to allelic bias *within white-striped birds only*. So, to be clear, we are not measuring allelic bias in tan-striped birds (this is, as noted, not possible). Furthermore, it is not a given that genes that are more highly expressed in tan individuals would show ZAL2 bias, *per se*. Genes that show differential expression, but not allelic bias, are likely showing differences in *trans*-regulation of the gene. Allelic bias, however, points specifically to *cis*-regulatory differences driving differential expression of the gene. Figure 4b is confirming with a larger sample size a finding that was presented in Sun et al., 2016. We have attempted to clarify these points in the text (see lines 389-392, 396-399, 402-403, and 437-439).

Other comments:– l. 243 Sentence hard to follow "We found evidence that allelic bias in gene expression was associated with the decile rank of the per-base number of non-synonymous fixed differences". Could this be assessed without using deciles: bias in gene expression vs. per-base number of non-synonymous fixed differences.

We instead now present the rate of fixed differences versus the decile rank in Figure 2 – supp. 1 and in the Results in the text on lines 245-250. This does not change the conclusions.

– l. 288. Please, provide an intuitive explanation of the D-statistic and what it means here. Difficult to see anything in Figure 3a.

Added “which tests for genomic regions that are discordant with the species tree” on lines 302-304.

– l. 425 Reference 63 corresponds to a web page with software. It is not clear what this reference means here. Please, check references to make sure that there are no errors (for example, see author list in reference 64).

We have added an additional reference for the H-scan software and have corrected the numbering errors for the citations. Other errors in the references have been corrected.

Reviewer #2 (Recommendations for the authors):Line 235: "whether degeneration … resulted in … reduced expression of the ZAL2m allele. " – ZAL2m is a whole supergene (or chromosome) variant. It is not an allele. Probably the following is better: "reduced expression of ZAL2m alleles". Throughout the manuscript, I don't think the authors should say "ZAL2m allele" when referring to the supergene variant. "Alleles carried by the Zal2m supergene variant" could be appropriate.

Thank you for this suggestion. We have made changes throughout.

Figure 4b: How many genes were there with the different types of bias? For genes with a higher expression in Tan than white, the authors suggest that the allelic bias towards the tan allele is a sign of adaptation. But it could simply be that the bias is the result of the ZAL2m allele having degenerated (similar to https://elifesciences.org/articles/55862 ). Conversely, while some higher expression of Zal2m alleles in white birds could be adaptive, some of it can also be that a transposon jumped in front of the promoter and leads to maladaptive high gene expression. We need to be careful with adaptationist interpretations.

These are good points. Genes showing ZAL2 versus ZAL2^m^ allelic bias are listed in Supp. Table 6. We have added additional caveats and alternative interpretations in the discussion on lines 538541.

Line 354 – finding extended homozygosity runs near the end of the chromosome arm could be expected from lower recombination in this region?

In the revised manuscript we used *Fasteprr* to infer the population recombination parameter, rho, for each genome assembly contig (Now Figure 4 – supp. 3). This analysis indicates that recombination rates are higher on the chromosome ends of ZAL2, but not on ZAL2^m^. Instead, recombination is very low throughout ZAL2^m^. Therefore, this pattern of elevated H-statistic is not caused by especially reduced recombination on the ends of the ZAL2^m^ chromosome. This has been clarified in the manuscript on lines 370-374.

Line 295-298: the authors check for introgression by building a tree from the region of Zal2m that is putatively balancing selection. A large amount of collapsed repetitive DNA in Zal2 can create misleading patterns in phylogenetic reconstruction attempts. This is due to the rapid turnover and jumping of repetitive elements (e.g., what looks like it is here in the reference genome may actually be from somewhere else, or could represent a hybrid genotype of several divergent repeat copies). Focusing on single-copy loci, such as exons of intact single-copy genes, should be preferred (see Stolle Nat Com2022 for one approach).

Unfortunately, our current data (mostly short read sequencing data) are insufficient to investigate structural variation at fine scale. We hope to follow up on this work using long-read sequencing in the future. To address this concern, we re-ran the phylogenetic analysis, this time reconstructing the phylogenetic tree of the ZAL2^m^ outlier region using only exons of the single copy orthologous genes. The results of this analysis, shown below and in Figure 3 – supp. 1, do not indicate an introgression event within ZAL2^m^. We have updated our Results to include mention of this analysis on lines 297-301.

Examination of allelic expression bias involves mapping to a genome where fixed differences between supergene variants have been masked. This is ok, but because the 2 supergene variants differ in the amount of genetic diversity they carry, and read mapping algorithms are sensitive to the number of differences between the reference sequence and the read, this would lead to a slight mapping bias. That creates a risk here. A more balanced control could have involved masking any genetic variation. However, I think that it is unlikely that doing so would fundamentally change the picture.

We have added this caveat to the Methods section where we describe the N-masking (lines 636-640).

Figure 1A shows that ZAL2m has higher overall genetic diversity than ZAL2, yet the text describing genetic diversity also consistently suggests the alternate pattern. For example, Ne derived from synonymous genic diversity is 10x lower in ZAL2m than ZAL2 (from line 222).Alternatively, the apparent conundrum could come from the use of different filtering criteria and the use of a linear reference sequence where repetitive regions are collapsed. The ZAL2m's genome sequence contains more difficult-to-assemble repetitive DNA, as evidenced by finding smaller scaffolds in Zal2m. This is in line with what is seen in the young fire ant Sb chromosomes (Pracana 2017 Mol Eco finding lower genetic diversity, Stolle 2018 finding more repetitive elements, and more variation in repetitive element content), as well as in some labile young Y chromosomes.

Figure 1a shows genetic diversity between tan-striped and white-striped birds, not between ZAL2 and ZAL2^m^ (this is actually shown in Figure 3A). Because all white-striped birds are heterozygous and all tan-striped birds are homozygous, there is higher genetic diversity in white-striped birds overall. Only in the haplotype-phased data, where reads are assigned to chromosomes, do we detect the pattern of overall reduced genetic diversity on ZAL2^m^. We have added a note to the caption of Figure 1 to make clear what is plotted in panel A.